# Regrowing the growth zone: metamorphosis kickstarts regeneration in the annelid *Capitella teleta*

Alicia A. Boyd* and Elaine C. Seaver‡

## ABSTRACT

The ability to regenerate can vary across the life history of an animal. We previously showed that *Capitella teleta*, an annelid worm, gains regenerative ability with age. Although larvae do not replace lost structures, juveniles and adults regenerate posteriorly following metamorphosis. To determine whether metamorphosis enables juveniles to regrow structures lost as larvae, we amputated *C. teleta* larvae, removing posterior segments, hindgut, posterior growth zone, anus and pygidium. Metamorphosis was then induced in these amputated larvae and they were reared as juveniles for 3, 7 or 14 days. New growth in juveniles was assessed by confocal microscopy, 5-ethynyl-2′-deoxyuridine incorporation, immunohistochemistry and nuclear staining. A posterior growth zone and new segments were observed by 7 days post-metamorphosis. Remodeling of the digestive system in pre-existing tissue was also observed. Additionally, re-amputation of juveniles resulted in regeneration of the posterior growth zone, segments and hindgut. Our results demonstrate that amputated *C. teleta* larvae can metamorphose into functional juveniles capable of growing new segments, suggesting that metamorphosis acts as a switch to enable regeneration of structures essential for growth. This study highlights the impact of metamorphosis on changes in developmental plasticity.

KEY WORDS: Regeneration, Metamorphosis, Developmental plasticity, *Capitella teleta*

## INTRODUCTION

Regeneration is the ability to replace lost tissues and structures. The regeneration capacity an animal can change substantially over the course of its life. Classic examples include amphibians, ascidians and echinoderms (Bothe and Fröbisch, 2024; Byrnes, 1904; Carnevali, 2006; Jeffery, 2015a,b; Phipps et al., 2020). This dynamic occurs in many animals with an indirect life cycle. Animals that undergo indirect development generate larval forms before metamorphosing into an adult form (Vervoort, 2011). For many marine invertebrates, metamorphosis is a pivotal developmental process marked by the presence of a rapid transition from functional, free-living larvae into juveniles that lose larval-specific structures and maintain the

Whitney Laboratory for Marine Bioscience, University of Florida, 9505 Ocean Shore Boulevard, St. Augustine, FL 32080-8610, USA.
*Present address: Living Systems Institute, University of Exeter, Stocker Road, Exeter EX4 4QD, UK.

‡Author for correspondence (seaver@whitney.ufl.edu)

A.A.B., 0000-0001-7107-5534; E.C.S., 0000-0002-5098-3899

functionality of adult structures generated as larvae and retained through metamorphosis (Hadfield, 2000; Hadfield et al., 2001).

As such, larvae of indirect developers can be surveyed for their regeneration potential and compared to regeneration abilities in adult stages. For example, in the anuran amphibian *Xenopus laevis*, larvae can regenerate multiple tissues and structures, including limbs. This ability is lost in adults after metamorphosis (Edwards-Faret et al., 2021; Gibbs et al., 2011; Harrison, 1898; Phipps et al., 2020; Slack et al., 2008). In contrast, the ascidian *Ciona intestinalis* cannot regenerate before metamorphosis (i.e. as embryos or larvae) but can as juveniles and young adults, although this ability is progressively lost with advanced age (Jeffery, 2015a,b). Other animals, such as some crinoids, display 'limited regeneration' as larvae, only to display robust regenerative ability following metamorphosis (Carnevali, 2006; Carnevali et al., 1993; Vickery et al., 2001). Thus, some animals lose regenerative ability after metamorphosis (e.g. amphibians), while others gain regenerative ability (e.g. ascidians and crinoids). For many indirect developers, it appears that metamorphosis is an event separating differences in regenerative ability.

*Capitella teleta* is an annelid worm that displays different regeneration potential before and after metamorphosis. *C. teleta* juveniles and adults have robust posterior regenerative ability but embryos show very limited regulation for blastomere loss (Amiel et al., 2013; Pernet et al., 2012). We recently showed that larvae have limited regenerative potential compared to juveniles (Boyd and Seaver, 2023); features characteristic of the initial stages of juvenile regeneration occur, namely wound healing, localized cellular proliferation at the wound site that coincides with stem cell marker expression, nerve extensions into the wound, and neural specification. However, neural differentiation or the replacement of structures was not observed in amputated larvae. It appears that *C. teleta* larvae can initiate the regeneration process but cannot successfully complete it, suggesting that *C. teleta* gains regenerative ability with age (Fig. 1A).

The presence of multiple morphological features of *C. teleta* can be leveraged to study this dynamic phenomenon. The segmented body of *C. teleta* has complex structures and organ systems, including a central nervous system, muscles and a regionalized digestive system. The ventral nerve cord (VNC) comprises a single ganglion in each segment, and bilateral pairs of peripheral nerves exit each ganglion (Meyer et al., 2015). Other segmentally repeated structures include the chaetae and segmental septae. Together, these structures enable the identification of individual segments and allow body regions to be distinguished.

Furthermore, juvenile and adult animals of *C. teleta* demonstrate indeterminate growth by continually generating new segments (Seaver and de Jong, 2021). New segments are sequentially generated by the posterior growth zone (pgz), a subterminal region located immediately anterior to the terminal pygidium (Fig. 1B,C) (Seaver et al., 2005). The pgz is characterized by proliferating, undifferentiated cells and expression of stem cell markers, such as

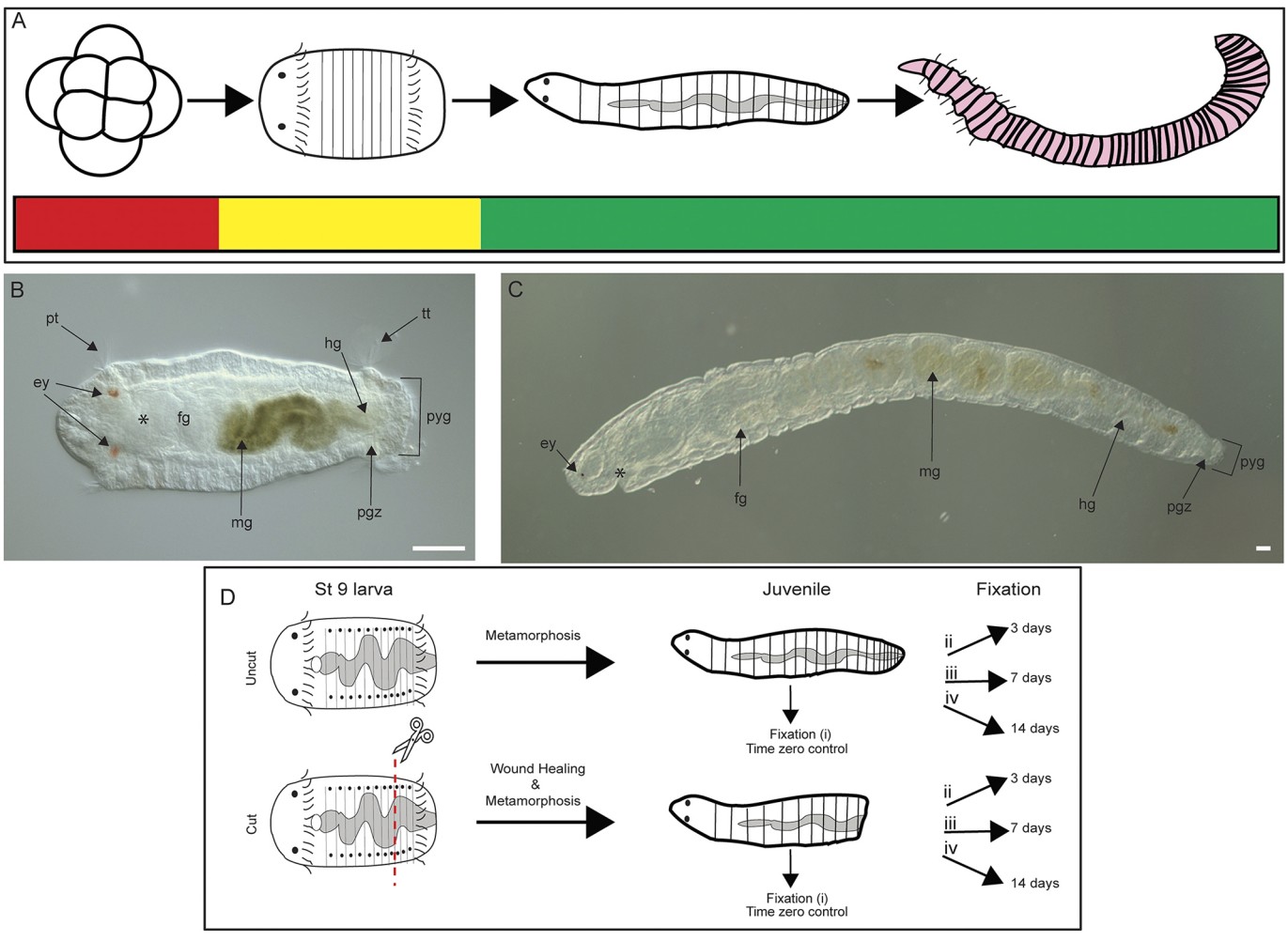

**Fig. 1. Evaluation of *Capitella teleta* regeneration following larval amputation and metamorphosis.** Anterior is to the left for all panels except the embryo in A. (A) *Capitella teleta* gains regeneration ability during development. Schematics show representative developmental stages (from left to right): embryo, larva, juvenile and adult. Red denotes lack of replacement of lost tissues (embryos), yellow denotes partial replacement (larvae) and green denotes complete regeneration (juveniles and adults). (B,C) Anatomical features of larva (stage 9; B) and two-week-old juvenile (C). Larva is in ventral view and juvenile is oriented in lateral view with ventral down. ey, eye; fg, foregut; hg, hindgut; mg, midgut; pgz, posterior growth zone; pt, prototroch; pyg, pygidium; tt, telotroch. Asterisk indicates the position of the mouth. Scale bars: 50 µm. (D) Schematic workflow of experiment. Amputation site denoted by scissors and red dashed line.

*vasa* and *piwi* (Dill and Seaver, 2008; Seaver and de Jong, 2021). As new cells are born through proliferation in the pgz, older cells are displaced anteriorly toward the more mature tissues and differentiate to form new segments (Zattara, 2020). Larvae generate approximately 13 segments before metamorphosis and have a pgz that expresses stem cell genes (Seaver, 2016). Juveniles and adults can fully regenerate posterior segments within 7 days and then indeterminately continue to grow (de Jong and Seaver, 2016). While amputated larvae cannot regenerate, they can successfully metamorphose into juveniles following exposure to a metamorphic cue (Boyd and Seaver, 2023). Together, these features provide an opportunity to study regeneration on either side of the metamorphic divide.

Here, we investigate whether amputated larvae can successfully replace their lost tissues as juveniles following metamorphosis, or whether regenerative ability is irrevocably lost. To address this, *C. teleta* larvae were amputated, induced to undergo metamorphosis and evaluated for regenerative potential as juveniles. Over 2 weeks, we characterized cellular proliferation and scored for formation of a pgz, new segments, and tissues such as digestive system subregions. Our aim was to improve our understanding of the relationship

between developmental maturation and regenerative potential. This study provides insights into the impact that metamorphosis can have on the shift of regenerative potential.

## RESULTS
### New segment growth
We investigated whether amputated larvae could grow new segments as juveniles. *C. teleta* larvae can be induced to undergo metamorphosis, as evidenced by loss of ciliary bands and initiation of burrowing behavior, with either vitamin B or marine sediment (Burns et al., 2014). We previously demonstrated that amputated, competent larvae successfully metamorphose into juveniles in less than 30 min when exposed to vitamin B (Boyd and Seaver, 2023). Following removal of the posterior third of the body by amputation and subsequent induction of metamorphosis, juveniles were scored for segment number at the following time points: immediately following metamorphosis (time zero) (Fig. 1D), 3 days post-metamorphosis (dpm), 7 dpm and 14 dpm (Fig. 1D). Animals were scored for segment number using the following three markers: presence of ganglia, paired peripheral nerves, and autofluorescence

of chaetae. Newly formed segments were located distal to the amputation plane and were smaller relative to pre-existing segments. The VNC was severed during amputation, resulting in a visible landmark for the precise location of the cut site.

Over 14 days, we observed the appearance of new tissue and posterior segment addition in both uncut and cut juvenile worms (Fig. 2A-Z). Uncut juveniles fixed at time zero had approximately 13-15 segments (Fig. 2A,C,E; $n$=54), whereas juveniles cut as larvae (JAL) had approximately 7-12 segments and lacked a pygidium (Fig. 2B,D,F; $n$=48; Table 1). By 3 dpm, uncut juveniles grew zero, one or two new segments (Fig. 2G,I,K; $n$=47). In contrast, the majority of JAL worms had only an unsegmented mass of tissue distal to the amputation site at 3 dpm, with a minority having one new segment (Fig. 2H,J,L; $n$=100). In the unsegmented new tissue mass of 3 dpm JAL worms, multiple neurites continuous with the connectives of the VNC were observed (Fig. 2L; $n$=55/65), with no discernable ganglia or peripheral nerves present (Fig. 2J,L). New segments, as indicated by presence of ganglia and peripheral nerves, were observed in most uncut and JAL worms by 7 dpm (Fig. 2M-R; $n$=64/66 and $n$=62/65, respectively). The width and length of these segments was visually smaller than those in pre-existing tissue. The VNC was present throughout the new tissue, with nerves spanning the pygidium (Fig. 2Q,R). In uncut worms, the total number of segments at 7 dpm ranged from 14 to 23, with between zero and eight

new segments ($n$=66). The number of total segments in JAL worms at 7 dpm ranged from 10 to 24, with between zero and six new segments ($n$=65). At 14 dpm, uncut worms had a range of 16-33 total segments, with approximately 3-18 new segments (Fig. 2S,U,W,Y; $n$=35). JAL worms at 14 dpm had 14-33 total segments, with approximately 7-21 new segments (Fig. 2T,V,X,Z; $n$=28). To determine whether cut worms regenerated new segments at a similar rate to uncut worms, we conducted an analysis of covariance from time zero until 14 dpm (Fig. 3A). These analyses showed a non-significant difference between uncut and cut worms across the 2 weeks ($P$=0.313). This indicates that JAL and uncut worms generate new segments at a similar rate. We also observed that the posterior-most tissue in 7 dpm JAL worms resembled a reformed pygidium, similar to that of uncut juveniles. In uncut juveniles, the pygidium was observed as an enlarged bulb of tissue posterior to a constriction of tissue and the pgz. Additionally, in the pygidium, the lateral-most nerves of the connectives flared laterally, away from the ventral midline (Fig. 2Q,W). In summary, animals amputated as larvae grow new segments as juveniles similarly to uncut juveniles and regenerate a pygidium-like structure.

### Regrowing the growth zone
The pgz is characterized by undifferentiated, proliferating cells and expression of stem cell markers in a band immediately anterior and adjacent to the pygidium (Fig. 4A; $n$=42/54) (Dill and

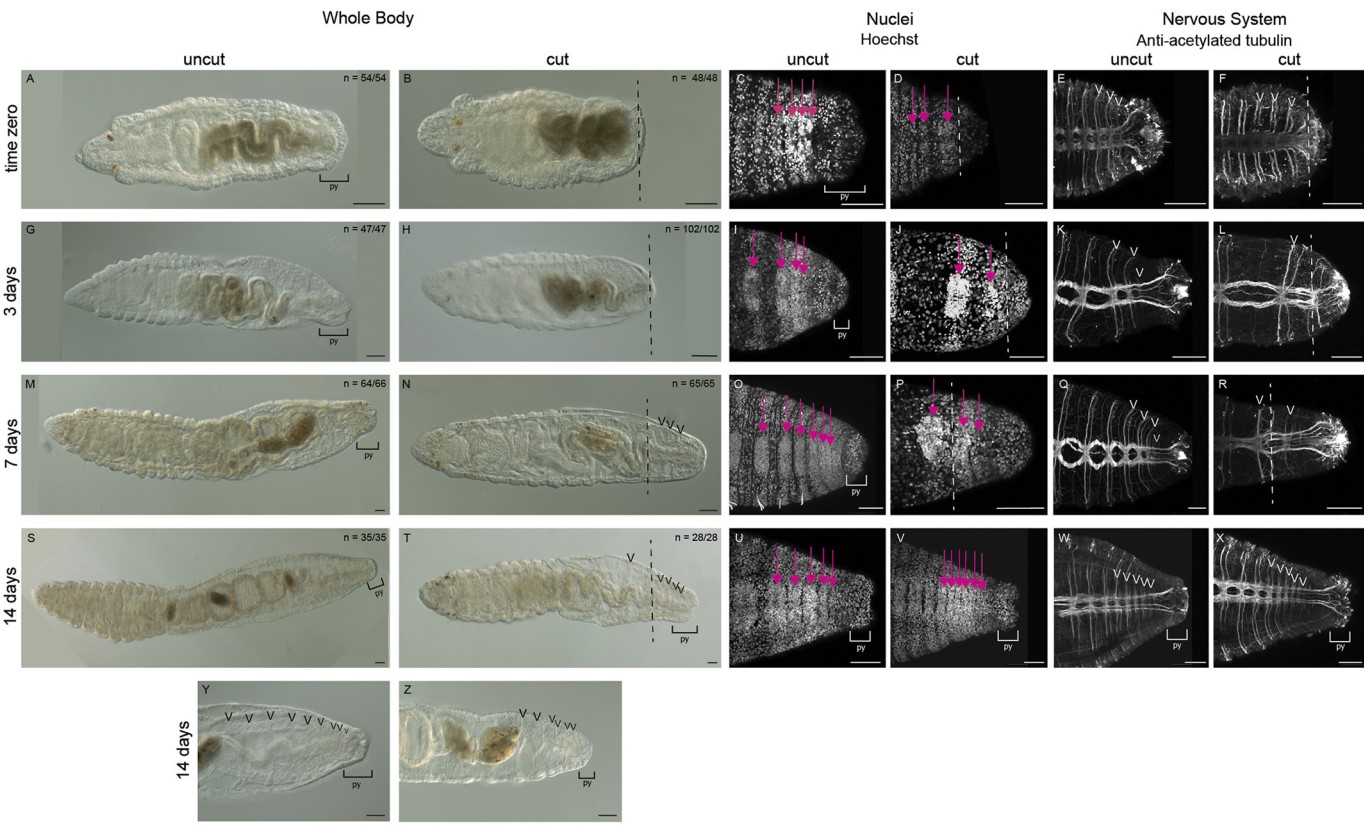

**Fig. 2. Juveniles amputated as larvae grow new segments.** All images are oriented in ventral view, with anterior to the left. (A-F) Controls fixed immediately following metamorphosis (time zero). (G-Z) Juveniles fixed at 3 dpm (G-L), 7 dpm (M-R) and 14 dpm (S-Z). (Y,Z) Magnified views of the posterior ends of 14 dpm juveniles. (A,B,G,H,M,N,S,T,Y,Z) Transmitted light images. (C-F,I-L,O-R,U-X) Confocal image stacks. (C,D,I,J,O,P,U,V) Nuclei are stained with Hoechst. Pink arrows indicate ganglia of the ventral nerve cord. (E,F,K,L,Q,R,W,X) Anti-acetylated tubulin antibody labeling of the nervous system. White arrowheads indicate pairs of peripheral nerves. Amputation site is represented by the dashed line. (V,X) In 14-day images, the amputation site is anterior to the region imaged, out of the field of view, and therefore not marked. A and C, J and L, P and R, V and X are the same animals, respectively. J and P are also shown in Fig. 4D,F, respectively. Bracket indicates the pygidium (py). $n$ denotes the number of animals scored that resemble the representative image, with each animal a biological replicate. Black arrowheads indicate individual segments by pointing to segmentally repeated septae. Two to four independent experiments were conducted for each time point. Scale bars: 50 μm.

**Table 1. Generation of new segments in uncut and cut animals**

| Treatment group | Range of segment number | Mean±s.d. | Variance | Range of new segments |
|---|---|---|---|---|
| Uncut time zero n=54 | 13-15 | 14±0.48 | 0.23 | 0 |
| Cut time zero n=48 | 7-12 | 9.6±1.25 | 1.56 | 0 |
| 3 day uncut n=47 | 14-17 | 15±0.60 | 0.36 | 0-2 |
| 3 day cut n=100 | 7-14 | 10.5±2.11 | 4.45 | 0-1 |
| 7 day uncut n=66 | 14-23 | 18±1.61 | 2.58 | 1-8 |
| 7 day cut n=65 | 10-24 | 14±2.78 | 7.70 | 0-6 |
| 14 day uncut n=35 | 16-33 | 24±3.90 | 15.23 | 3-18 |
| 14 day cut n=28 | 14-33 | 20±5.06 | 25.56 | 7-21 |

For each treatment group, the range of total segment number, mean±s.d., variance, and range of new segments generated during the experiment is reported. *n* denotes the number of animals scored for each treatment group, with each individual animal counted as a biological replicate.

Seaver, 2008). We examined 5-ethynyl-2′-deoxyuridine (EdU) incorporation and expression of the stem cell marker *vasa* as indicators of a pgz in cut and uncut worms. In amputated larvae, the pgz was removed, and EdU$^+$ cells were not detectable at the wound site when examined immediately after metamorphosis (Fig. 4B; *n*=48/48). At 3 dpm, uncut worms had a distinct subterminal band of EdU$^+$ cells, anterior to the pygidium (Fig. 4C; *n*=46/47). In 3 dpm cut worms, there were numerous EdU$^+$ cells distal to the cut site that spanned the new tissue from the cut site to the posterior end of the animal and were present in the ectoderm and mesoderm (Fig. 4D; *n*=95/102). By 7 dpm, a subterminal band of EdU$^+$ cells was present in both uncut (Fig. 4E; *n*=64/66) and cut (Fig. 4F;

*n*=63/63) worms, indicating that the pgz had reformed in cut animals. To provide additional support to the assertion that this band of EdU$^+$ cells was a pgz, we examined expression of the marker *vasa* in 7 dpm worms (Fig. 5) (Dill and Seaver, 2008). As expected, the expression domain of *vasa* was missing from the posterior end of time zero amputated animals, consistent with removal of the pgz during amputation (Fig. 5A; *n*=26/26). In uncut juveniles, *vasa* was expressed in the ectoderm and mesoderm in a subterminal domain (Fig. 5B; *n*=15/21). In 7 dpm JAL worms, there was novel subterminal *vasa* expression (Fig. 5C; *n*=17/18), confirming the re-establishment of the pgz in juveniles.

## Gut remodeling and regeneration

The digestive system of *C. teleta* is regionalized and broadly divided into foregut, midgut and hindgut (Fig. 6A-D′). These regions can be distinguished by differences in labeling patterns with an anti-acetylated tubulin antibody (Meyer et al., 2015). In the foregut and hindgut, the most prominent labeling included cilia in the digestive track lumen, whereas in the midgut a complex web of neurites of the enteric nervous system is the most visible feature (Fig. 6A-B,D′; *n*=47/47). Amputations in the midgut provide the advantage of being able to distinguish between midgut and hindgut identity in regenerating tissues.

From time zero until 14 dpm in uncut animals (Fig. 6E-H), a gradual, but identifiable delineation could be made between the midgut and ciliated hindgut. In time-zero JAL worms, posterior ciliation in the hindgut was removed as the result of amputation in the midgut (Fig. 6I; *n*=48/48) and contrasts with time-zero uncut animals (Fig. 6E; *n*=52/54). By 3 dpm, a new ciliated hindgut was observed in JAL animals (Fig. 6J; *n*=91/105), although it was shorter relative to the uncut controls (Fig. 6F; *n*=47/47). The new ciliation in the gut lumen of cut animals appeared in pre-existing tissue proximal to the cut site as well as in the newly regenerated tissue distal to the cut site (Fig. 6J). Presence of gut ciliation on either side of the amputation

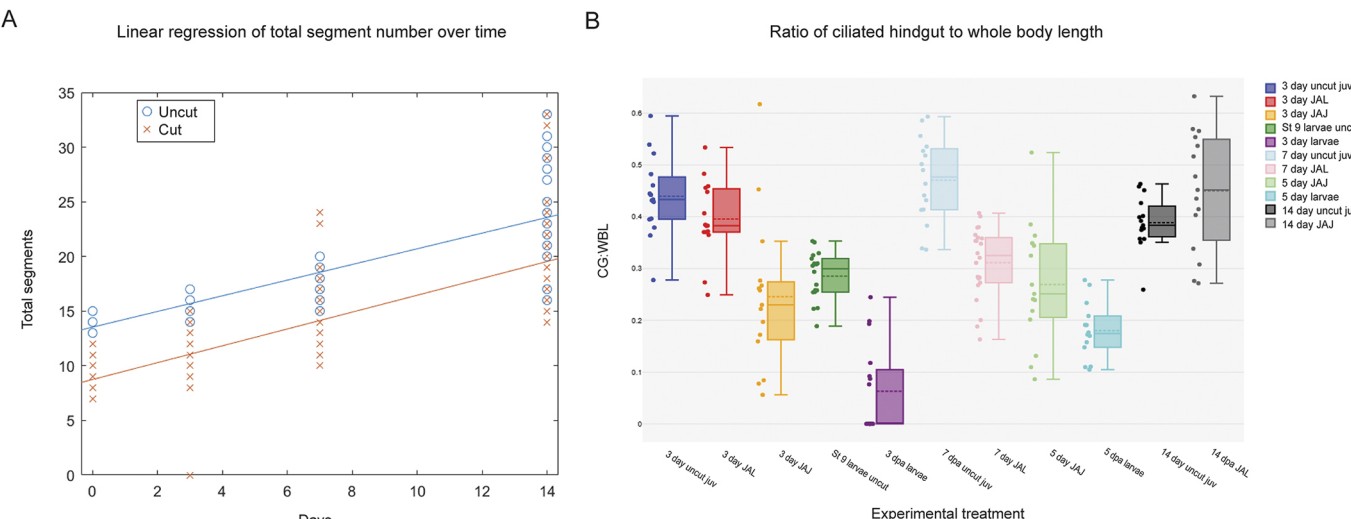

**Fig. 3. Quantitative data for new segment growth and ciliated hindgut:whole-body length ratio across time.** (A) ANOCOVA linear regression of total segment number across time for uncut and cut worms. *y*-axis is total segment number; *x*-axis is the number of days. Blue circles represent uncut worms. Red crosses represent cut worms. (B) Box-and-whisker plot showing the proportion of the body length composed of ciliated hindgut for each experimental group. *y*-axis plots the ciliated hindgut:whole body ratio. CG, ciliated gut; WL, whole-body length. *x*-axis includes each experimental group. Uncut and cut animals for each experimental group are represented by the following colors: 3-day uncut juvenile, dark blue; 3-day JAL, red; 3-day JAJ, yellow; uncut stage 9 larvae, green; 3 dpa larvae, purple; 7-day uncut juvenile, light blue; 7-day JAL, pink; 5-day JAJ, light green; 5-dpa larvae, teal; 14-day uncut juvenile, black; and 14-day JAL, gray. JAL, juveniles amputated as larvae; JAJ, juveniles amputated as juveniles; juv, juveniles. Each ratio is represented by a dot on the plot, next to the associated box-and-whisker in matching color. The box limits are the quartiles calculated by Statistics Kingdom linear option. The whiskers extend to the data point within 1.5 times the IQR of the upper and lower quartile. Within each box, dotted horizontal lines indicate the mean and solid lines indicate the median.

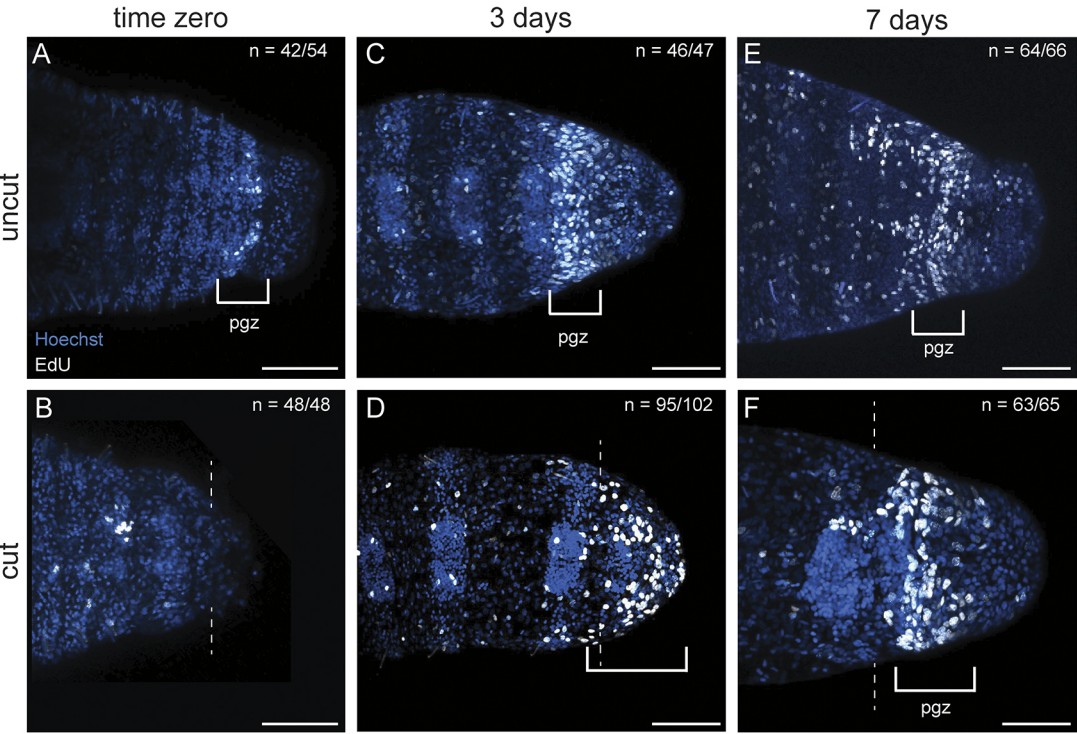

**Fig. 4. Posterior growth zone is re-established in amputees by 7 dpm.** Blue is Hoechst staining; white nuclei mark EdU incorporation. All images are oriented in ventral view, with anterior to the left. (A,C,E) Uncut juvenile controls. (B,D,F) Experimental juveniles cut as larvae, metamorphosed and raised for various time intervals. Animals were fixed immediately following metamorphosis (A,B) or at 3 dpm (C,D) or 7 dpm (E,F). D and F are also shown in Fig. 2J,P, respectively. White brackets indicate localized area of proliferating cells. Dashed line indicates the amputation site. *n* denotes the number of animals scored that are similar to the image represented, with each animal as a biological replicate. Three independent experiments were conducted for each time point/treatment. pgz, posterior growth zone. Scale bars: 50 μm.

plane persisted over 14 days (Fig. 6J-L). As illustrated in whole-body images, the anterior boundary of ciliation in JAL animals at 3 dpm was in a segment (e.g. segment 11; Fig. 6C,D) that in an uncut animal had midgut identity and therefore lacked cilia (Fig. 6A,B). This result indicates a remodeling of pre-existing tissue.

To better characterize the replacement of hindgut in JAL worms, the lengths of ciliated hindgut and whole body were measured. From these measurements, the proportion of the body constituted by the ciliated hindgut was calculated (Table S1) and compared between cut and uncut animals of similar ages following

metamorphosis (Fig. 3B, Tables 2 and 3). These proportions were also calculated in the context of successful posterior regeneration in juveniles amputated as juveniles (JAJ).

The difference in the ratios of the ciliated hindgut length to whole-body length between uncut 14 dpm juveniles and 2-week-old JAJ worms was statistically significant when evaluated at 3 or 5 days post-amputation (dpa) (Fig. 3B, Tables 2 and 3). This comparison was used as a framework for understanding how the hindgut regenerates in JAL worms compared with uncut juveniles of the same age. We hypothesized that the length ratios would

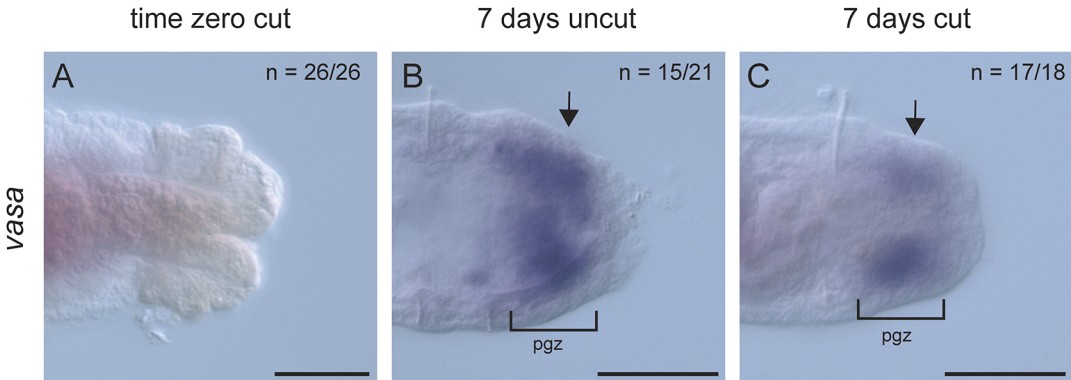

**Fig. 5. The posterior growth zone marker *vasa* is expressed in amputees by 7 dpm.** All images are of the posterior end of juveniles oriented in ventral view, with anterior to the left. (A) Cut at time zero, immediately following amputation/metamorphosis. (B) Uncut 7-day-old juvenile control. (C) 7-day-old juvenile that was cut as a larva. Black brackets and arrows indicate subterminal domain of purple precipitate indicative of *vasa* expression. This expression pattern is typical of the posterior growth zone (compare B and C) (Dill and Seaver, 2008). *n* denotes the number of animals scored similar to the image represented, with each animal as a biological replicate. Two independent experiments were performed. pgz, posterior growth zone. Scale bars: 50 μm.

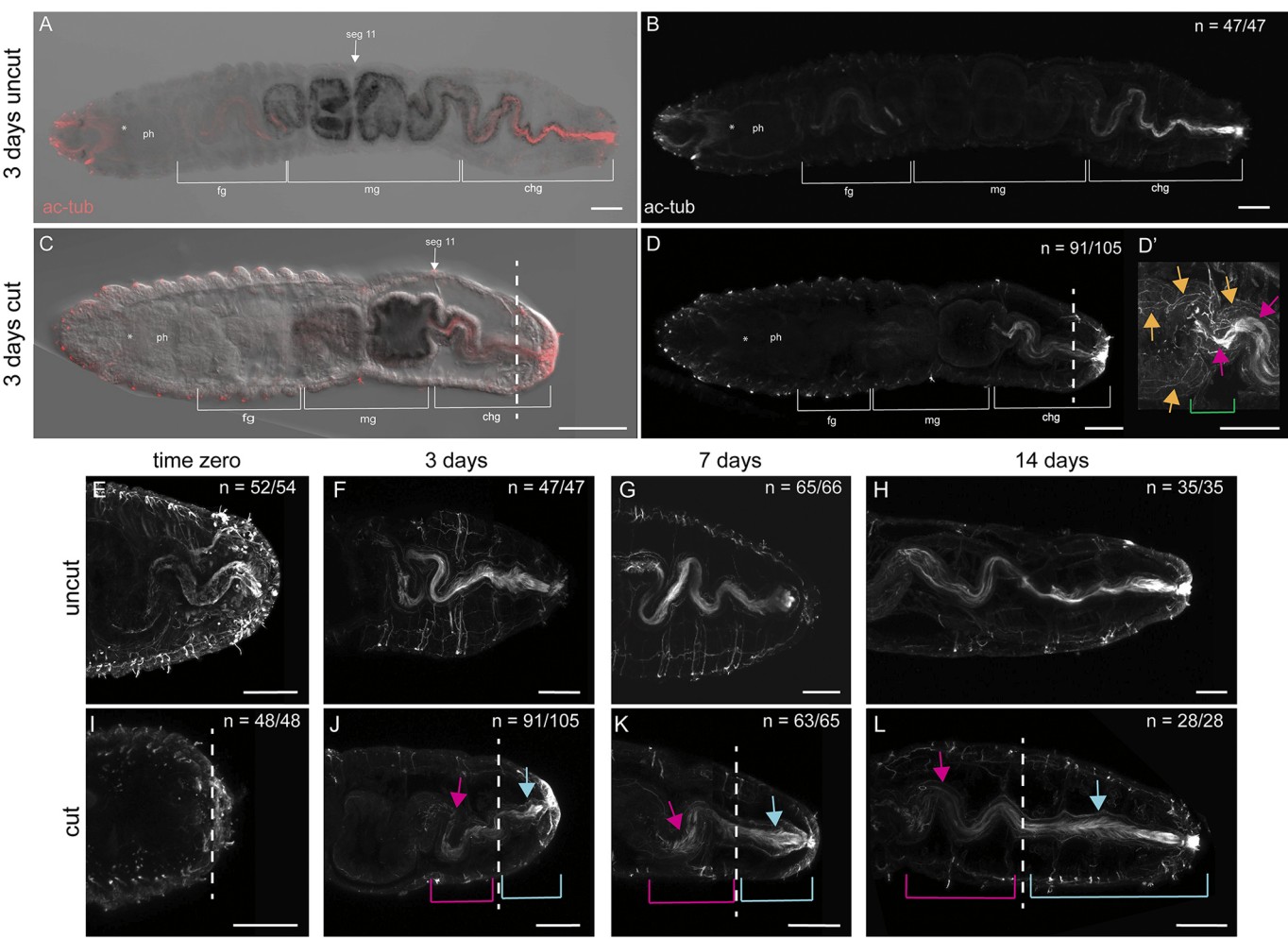

**Fig. 6. Posterior regeneration includes generation of ciliated hindgut in new tissue and remodeling of pre-existing gut.** Ciliation is marked with anti-acetylated tubulin antibody and appears red in A,C and white in B,D-L. All images are oriented in ventral view, with anterior to the left. (A,B,E-H) Uncut juveniles. (C-D′,I-L) Juveniles amputated as larvae. A and B show the same animal and are also seen in Fig. 7A,A′. C-D′ show the same animal. White brackets in A-D indicate subregions of the digestive system. Green bracket in D′ indicates the transition of the midgut to the hindgut. The white arrows in A and C indicate the location of segment 11. The yellow arrows in D′ indicate enteric nerves of the midgut. Pink arrows in D′ indicate ciliation in the gut lumen. Juveniles were fixed either immediately following metamorphosis (E,I) (time-zero animals), 72 h post-metamorphosis (F,J), 7 dpm (G,K) or 14 dpm (H,L). F is also seen in Fig. 4C. H, J and L are also shown in Fig. 7B-D′, respectively. Pink arrows and brackets indicate ciliated hindgut in pre-existing tissue. Teal arrows and brackets indicate ciliated hindgut in regenerated tissue. White dashed line indicates the approximate amputation site. Asterisks indicate position of the mouth. *n* denotes the number of animals scored that are similar to the image represented, with each animal as a biological replicate. chg, ciliated hindgut; fg, foregut; mg, midgut; ph, pharynx. At least three independent experiments were performed for each group. Scale bars: 50 μm.

be statistically different between uncut and JAL juveniles at 3 dpm and 7 dpm, with the hindgut gradually recovering normal proportional length by 14 dpm. Surprisingly, we found that by 3 dpm, when ciliation first appeared in the lumen of cut animals, the total proportion of the body made up of ciliated hindgut was similar between uncut and cut animals and the difference was non-significant (Table 3). As expected, the difference in the ratio of hindgut length to whole-body length between cut and uncut animals was statistically significant at 7 days (Fig. 3B, Table 3); indicating that the length of the hindgut is proportionally shorter in cut animals than in uncut animals. The hindgut of cut animals then continued to elongate, and by 14 dpm the proportion of the hindgut to whole-body length was similar and non-significant between cut and uncut animals (Figs 6H,L and 3B, Table 3). The similarity of hindgut length observed at 14 dpm indicates a replacement of hindgut in cut animals resulting from a combination of remodeling of pre-existing tissue and generation of new segments to produce a new, continuous ciliated hindgut.

To evaluate whether hindgut regeneration occurs similarly between JAJ and JAL worms, the ratios of hindgut length to whole-body length were compared between these groups. There was a significant difference between JAJ at 3 dpa and JAL at 3 dpm, but not between JAJ at 5 dpa and JAL at 7 dpm (Fig. 3B, Table 3). These results indicate that regenerating juveniles recover the correct proportionality of the hindgut differently, potentially at different rates, depending on the developmental stage at which the initial amputation occurs.

Since the hindgut continues to elongate in cut and uncut animals as the juveniles grow, we looked for dividing cells in the hindgut epithelium. In uncut juveniles, EdU$^+$ cells were present in the epithelial lining of the hindgut at 3 dpm (Fig. 7A,A′). By 14 dpm, the number of EdU$^+$ cells visually decreased in the hindgut epithelium (Fig. 7B,B′). In contrast, other surrounding tissues (e.g. pgz) did not show the same apparent decrease in EdU incorporation. In cut animals at 3 dpm, EdU$^+$ cells were also present in the hindgut epithelium (Fig. 7C,C′). However, EdU$^+$ cells persisted throughout the hindgut epithelium for 2 weeks (Fig. 7D,D′); this result contrasts

**Table 2. Variation in hindgut growth as determined by the ratio of ciliated hindgut length:whole-body length**

| Experimental type | Treatment group | Range of CG:WBL ratios | Mean±s.d. | Variance |
|---|---|---|---|---|
| Juveniles amputated as larvae (JAL) | 3 day uncut $n$=15 | 0.28-0.59 | 044±0.08 | 0.006 |
| | 3 day cut $n$=15 | 0.25-0.53 | 0.39±0.74 | 0.005 |
| | 7 day uncut $n$=16 | 0.34-0.59 | 0.47±0.08 | 0.007 |
| | 7 day cut $n$=22 | 0.16-0.41 | 0.31±0.07 | 0.005 |
| | 14 day uncut $n$=15 | 0.26-0.46 | 0.39±0.05 | 0.003 |
| | 14 day cut $n$=15 | 0.27-0.63 | 0.45±0.11 | 0.013 |
| Juveniles amputated as juveniles (JAJ) | 3 day cut $n$=15 | 0.06-0.62 | 0.25±0.15 | 0.02 |
| | 5 day cut $n$=15 | 0.09-0.52 | 0.27±0.12 | 0.01 |
| Amputated larvae | St 9 uncut $n$=18 | 0.19-0.35 | 0.29±0.05 | 0.002 |
| | 3 day cut $n$=16 | 0.0-0.24 | 0.06±0.09 | 0.007 |
| | 5 day cut $n$=14 | 0.10-0.28 | 0.18±0.05 | 0.003 |

CG:WBL, ciliated hindgut length:whole-body length.
For each treatment group, the range of ciliated hindgut: whole body length ratios, mean±s.d., and variance is reported. $n$ denotes the number of animals scored for each treatment group, with each individual animal counted as a biological replicate.

with uncut animals of the same age. Across all samples, an aggregation of EdU$^+$ nuclei occurred in the gut epithelium at the midgut/hindgut transition. The presence of dividing cells in the growing hindgut epithelium is consistent with a local cellular origin of new tissue.

To better understand whether remodeling of the hindgut in pre-existing tissue is unique to juveniles, we also analyzed amputated larvae for reappearance of ciliation. In stage 6 larvae, the digestive system is still maturing and therefore does not have ciliation in the posterior portion of the digestive tract (Fig. 8A). The area in which the hindgut normally develops was removed during amputation (Fig. 8B). In late-stage larvae (stage 9, uncut), the ciliated hindgut was now present in the posterior end of the trunk and pygidium (Fig. 8C,C′; $n$=13/13). By 5 dpa, half of the cut animals had gut

**Table 3. $t$-test statistical analysis of hindgut length:whole-body length ratio between treatment groups**

| Experimental type | Groups compared | $P$-value | Significance ($P$<0.05) |
|---|---|---|---|
| Juveniles amputated as larvae (JAL) | 3 day uncut and 3 dpa cut | 0.1237 | Non-significant |
| | 7 day uncut and 7 dpa cut | 1.334e−7, | Significant |
| | 14 day uncut and 14 dpa cut | 0.06792 | Non-significant |
| Juveniles amputated as juveniles (JAJ) | 14 day uncut and 3 dpa cut | 0.001315 | Significant |
| | 14 day uncut and 5 dpa cut | 0.001091 | Significant |
| Amputated larvae | St 9 uncut and 5 dpa larvae | 0.000001801 | Significant |
| Cross group comparison | JAJ 3 dpa and JAL 3 dpa | 0.00142 | Significant |
| | JAJ 5 dpa and JAL 7 dpa | 0.1767 | Non-significant |

For each analysis, the $P$-value is listed with its concordant statistical relationship as determined by an alpha value of 0.05.

ciliation and there was a statistically significant difference in length compared with uncut stage 9 larvae (Table 3, Fig. 8D,D′; $n$=9/16). This ciliated region was shorter than that observed in uncut, age-matched larvae and was present in the pre-existing tissue (Fig. 8D,D′). Therefore, despite removal by amputation of hindgut precursors, these results show that remodeling of pre-existing tissue occurs in amputated larvae, albeit more slowly and to a lesser degree than in amputated juveniles.

### Re-amputation experiment

To evaluate whether JAL worms respond to amputation in the same manner as unmanipulated juveniles, we amputated two groups of 2-week-old juveniles at segment 10 (Fig. 9A). One group was previously amputated as larvae, metamorphosed, and reared for 2 weeks before being amputated again as juveniles. The control group was amputated once as 2-week-old juveniles. Worms that were amputated as larvae and again as juveniles are referred to as 'double-cut' and the worms cut once as juveniles are referred to as 'single-cut' controls. The animals were scored for formation of new segments at 7 dpa. The presence and number of segments were identified using the same characteristic markers as in the previous experiment, i.e. newly formed ganglia and pairs of peripheral nerves distal to the cut site; 7 dpa is sufficient time for regeneration of a pgz and multiple new segments in 2-week-old juveniles, thus both features could be scored.

Both single-cut and double-cut worms regenerated after 7 days. Single-cut worms grew between zero and 13 new segments, with an average of five new segments. The total number of segments ranged from 10 to 23 segments, with an average of 15 total segments (Fig. 9B,D; $n$=44). Double-cut worms grew between one and 17 new segments, with an average of seven new segments. The total number of segments in double-cut worms ranged from 11 to 27, with an average of 17 total segments (Fig. 9C,E; $n$=27). The difference in the number of segments generated post-amputation between single-cut and double-cut worms was statistically significant ($t$-test, $P$=0.04205). Based on the number of new segments observed in both

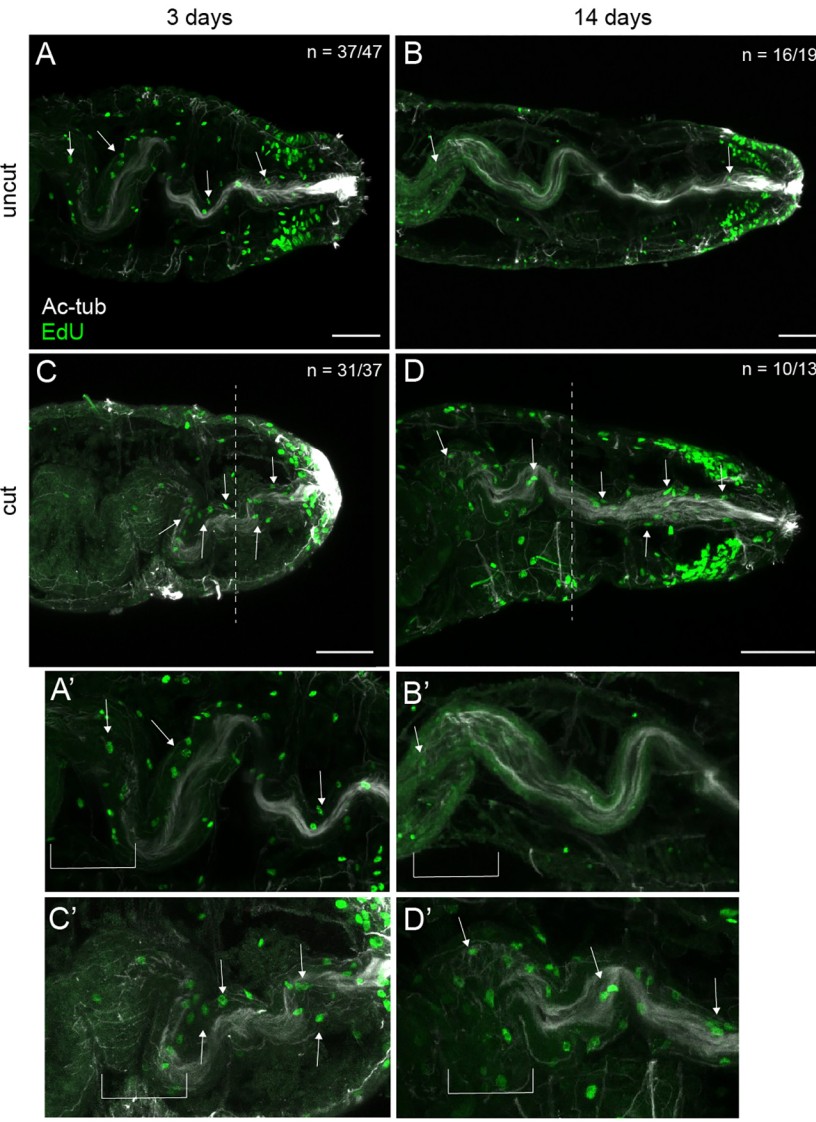

**Fig. 7. Elevated proliferating cells in the regenerating hindgut epithelium.** EdU-positive nuclei are green. Ciliated hindgut is visible with anti-acetylated tubulin antibody labeling (white). All images are oriented in ventral view, with anterior to the left. (A,B) Uncut animals. (C,D) Juveniles cut as larvae. (A,C) 3 dpm juveniles. (B,D) 14 dpm juveniles. A′-D′ show magnified images of A-D, respectively. A and A′ were also seen in Fig. 6A,B. B-D′ are from Fig. 6H,J,L, respectively. White arrows indicate EdU-positive cells in the hindgut. White dashed lines denote approximate amputation site. White brackets indicate the region in which the midgut transitions to the hindgut. $n$ denotes the number of animals scored that resemble the representative image, with each animal as a biological replicate. At least two independent experiments were conducted for each treatment. Scale bars: 50 µm.

groups, it appears that double-cut worms generate more segments over 7 days than single-cut worms. A new pgz was visualized by the presence of a subterminal band of EdU$^+$ nuclei in single-cut (Fig. 9F; $n$=40/44) and double-cut (Fig. 9G; $n$=26/27) juveniles. Ciliated hindguts were present in both single-cut ($n$=41/44) and double-cut ($n$=27/27) worms. Taken together, double-cut worms successfully regenerated in a similar, albeit faster, manner than single-cut worms.

## DISCUSSION
### *C. teleta* replaces lost tissues only after metamorphosis
In this study, we demonstrate that metamorphosis is pivotal for *C. teleta* to successfully regenerate posteriorly. We investigated whether regenerative ability was inherent to specific life stages or could be influenced, even hindered, by tissue removal during a previous life stage. *C. teleta* were amputated during a regenerative incompetent stage, i.e. larva, and evaluated for the replacement of lost structures during a regenerative-competent stage, i.e. juvenile. We found that juveniles derived from amputated larvae replaced the pgz, nerves and digestive system, and generated new, fully patterned segments following metamorphic induction. These new tissues appear to be functional; the presence of fecal pellets in the digestive tract is consistent with restoration of a functional digestive

system (A.A.B., personal observation), and the continued addition of new segments is indicative of a functional pgz. Moreover, when amputated a second time as juveniles, the worms once again underwent successful posterior regeneration. It is notable that the second amputation stimulus in double-cut worms appears to increase the rate of segment generation when compared with single-cut juveniles. With these observations, it is apparent that (1) amputation during larval stages does not hinder the regeneration ability of juveniles, (2) following metamorphosis, the animal is capable of detecting missing tissues and structures, and (3) regeneration can be initiated when temporally removed from conventional wound-related triggers.

The ability to fully regenerate in *C. teleta* appears to be enabled by metamorphosis. Prior to metamorphosis, amputated larvae displayed the beginning stages of posterior regeneration, including wound healing, localized cell proliferation and stem cell marker expression at the wound site, and neural extensions into the wound (Boyd and Seaver, 2023). In most cases, lost structures were not replaced in larvae, i.e. ciliary bands, pgz, pygidium, etc. Similarly, a previous study showed that only 13% of *C. teleta* larvae exhibited some degree of germline precursor regeneration in response to a targeted deletion of the germline precursor in embryos (Dannenberg

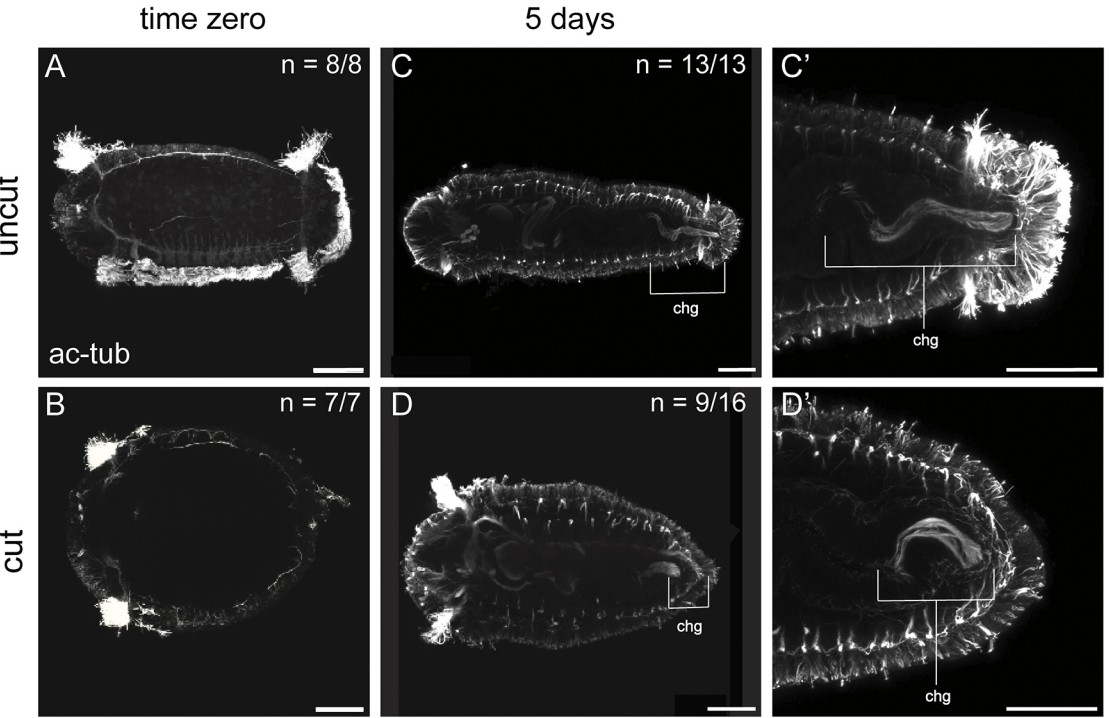

**Fig. 8. Amputated larvae develop ciliated hindgut.** All images are larvae, labeled with an anti-acetylated tubulin antibody (white), and are oriented in ventral view, with anterior to the left. (A,B) Larvae fixed at stage 6 following amputation (B) or as uncut controls (A) (time zero). (B) Cut animals wound healed prior to fixation. (C,C′) Stage 9 uncut larvae. (D,D′) Larvae 5 dpa. C′ and D′ show enlarged views of C and D, respectively. Brackets indicate the ciliated hindgut. *n* denotes the number of animals scored that resemble the representative image, with each animal as a biological replicate. chg, ciliated hindgut. At least two independent experiments were conducted for each time point. Scale bars: 50 μm.

and Seaver, 2018). Thus, limited regenerative ability is observed prior to metamorphosis. In contrast, the ability to regenerate posterior structures appears to dramatically increase after metamorphosis. For example, following deletion of germline precursors in the embryo, all 2-week-old juveniles resulting from this manipulation successfully regenerated germline cells (Dannenberg and Seaver, 2018). In our study, amputated larvae can successfully replace the pygidium and pgz after metamorphosing into juveniles. These observations demonstrate that successful posterior regeneration is limited in pre-metamorphic larvae and complete in post-metamorphic worms.

There may be physiological and molecular factors that contribute to the differential regenerative potential observed on either side of the metamorphic divide. Perhaps the most prominent physiological change between larval and juvenile stages is the switch in nutritional source from yolk-derived to active feeding. *C. teleta* larvae are non-feeding and are sustained through metamorphosis from yolk stores. The yolk is gradually consumed until stage 9, at which point the digestive system has matured and the yolk has been metabolized. *C. teleta* juveniles initiate eating within 1-2 h after metamorphosis (Muhl, 2022). This immediate nutritional input could provide the necessary energy to complete the remaining stages of regeneration. Furthermore, informal observations indicate that the extent of regeneration in *C. teleta* juveniles depends on nutritional input (E.C.S. and A.A.B., personal observations). The connection between nutrition and regenerative ability has also been observed in *Xenopus tropicalis* larvae. As tadpoles mature and deplete their yolk stores, their regenerative competence diminishes, yet excess feeding of these tadpoles can prolong regenerative success (Williams et al., 2021).

The change in regenerative competency may also be due to molecular differences between the two life stages. This could manifest as an absence of a molecular signal in larvae that is only present post-

metamorphosis, or an inability/immaturity of larval tissues to respond to a molecular signal that would otherwise promote continuation of the regenerative process. Although extremely rare (i.e. in <10%), some amputated larvae regenerate a telotroch and/or pygidium (Boyd and Seaver, 2023). Similarly, we observed some amputated larvae at 5 dpa that had remodeled pre-existing tissues into ciliated hindgut (Fig. 7; *n*=9/16). These observations reveal that, pre-metamorphosis, there is potential in *C. teleta* to progress past the initial stages of the regenerative program. Further studies are needed to determine whether these further regenerated larvae have molecular or nutritional advantages over their less-regenerative counterparts. We hypothesize that the change in *C. teleta* regenerative competency between larvae and juveniles is the result of nutritional and molecular shifts that coincide with metamorphosis.

Our experimental manipulations uncouple the events of regeneration by inducing metamorphosis directly after the amputation stimulus, thereby temporally separating the surgical amputation stimulus from the initiation and completion of regeneration via induction of metamorphosis. The molecular and physiological signals initiated by amputation likely persist through the process of metamorphosis. An alternative hypothesis is that only after metamorphosis can the animal detect missing tissue, and it is the missing tissue that serves as the primary signal to initiate regeneration in juveniles.

Our observations in *C. teleta* represent just one example of the varied regenerative potential observed in annelids. In posterior regeneration of many annelids, the pgz and pygidium are the first structures to regenerate (Boilly et al., 2024). Our results in *C. teleta* are similar; we observe that the pgz and pygidium are formed in JAJ by 4 dpa and in JAL between 3 and 7 dpa. The results of our re-amputation experiment in juveniles support previous observations that *C. teleta* adults are resilient to repeated amputation

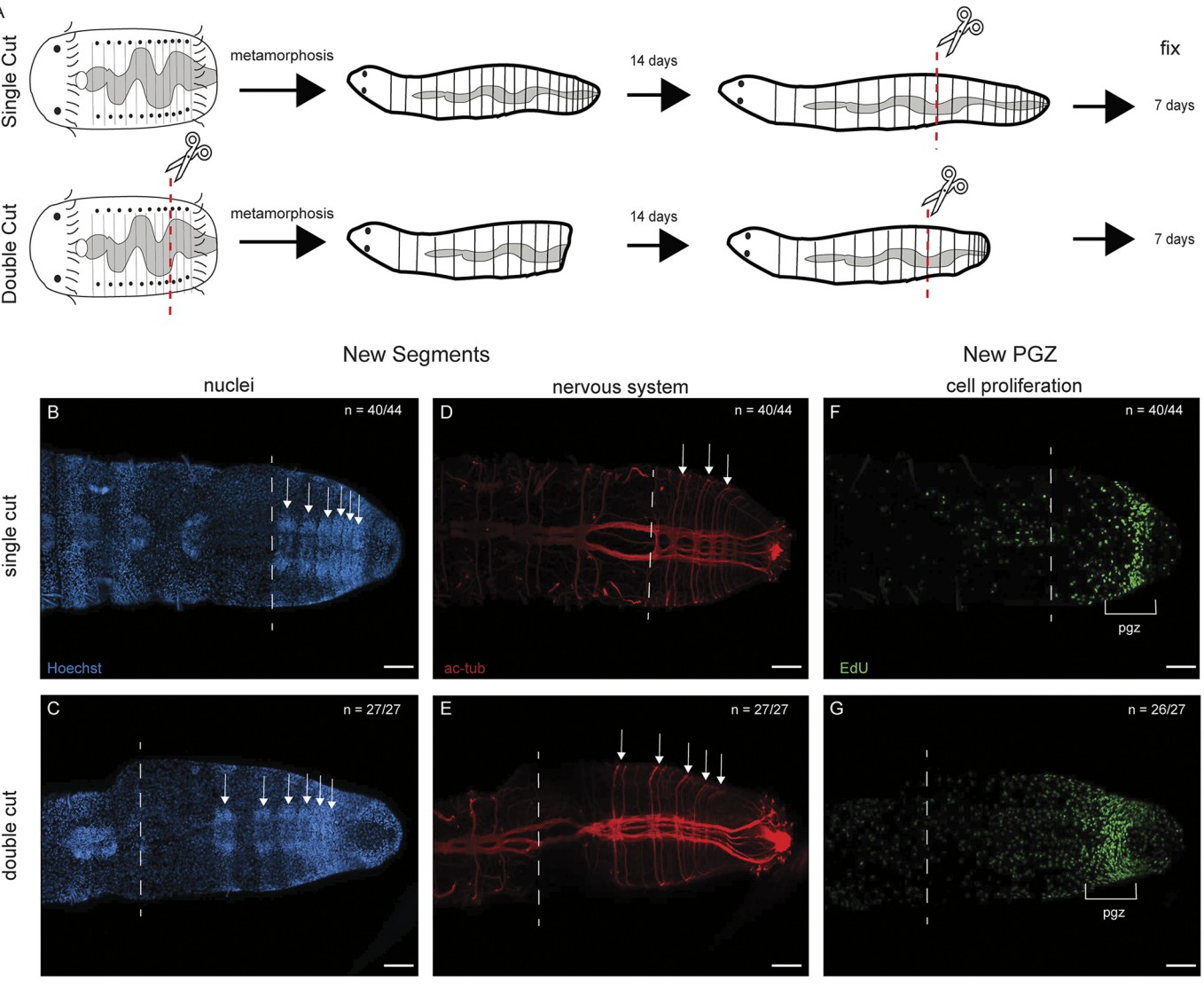

**Fig. 9. Successful regeneration following sequential amputations.** All images are oriented in ventral view, with anterior to the left. (A) Schematics showing experimental design of the re-amputation experiment. 'Single-cut' denotes controls amputated only as juveniles. 'Double-cut' denotes animals amputated as larvae and again as juveniles. Amputation site denoted by scissors and red dashed line. (B-G) Juveniles 7 dpm. (B,C) Nuclear staining (blue) highlights the segmental ganglia (arrows). (D,E) Anti-acetylated tubulin labeling (red) shows paired peripheral nerves and ganglia of individual segments. (F,G) EdU incorporation (green) showing the position of the posterior growth zone (pgz). *n* denotes the number of animals scored that resemble the representative image, with each animal as a biological replicate. White arrows denote individual segments. The dashed line indicates the amputation site. The bracket marks the boundaries of the pgz. Two independent experiments were conducted for each treatment group. Scale bars: 50 μm.

(Hill et al., 1986). The rate of new segments generated by juveniles that were amputated as larvae is similar to that of uncut animals at 1 and 2 weeks post-amputation, suggesting that metamorphosis resets the regeneration program, independently of the number of amputation events. We also observed that, while the rate of growth between uncut and cut JAL worms was similar, the double-cut worms regenerated more segments than single-cut worms. Other annelids, such as *Syllis amica* and *Platynereis dumerilii*, regenerate new segments at a significantly faster rate immediately following amputation (i.e. within the first 10 days) than in the following weeks (Boilly et al., 2024; Metzger and Özpolat, 2024). In *S. amica*, the end of the initial regenerative response is marked by a return in growth rate to that of unamputated worms (Boilly et al., 2024). In summary, there are both similarities and differences in the specifics of posterior regeneration between our observations in *C. teleta* and those from other adult annelids.

## Remodeling the digestive tract

We observed hindgut regeneration distal to the cut site in both amputated larvae and JAL that was independent of successful regeneration in other tissues. Notably, we also observed hindgut tissue remodeling of pre-existing tissue proximal to the cut site in JAL animals within 3 dpa in regions that previously exhibited midgut characters. This remodeling occurred prior to new segment formation distal to the amputation site. Furthermore, statistical analyses of the ratio of ciliated hindgut length to whole-body length at 3 dpm between uncut and cut animals indicate that the hindgut length of cut animals is not significantly different from that of uncut animals. In contrast, the ratio of hindgut to whole body in JAJ at 3 dpa is significantly different from that in uncut juveniles, indicating that JAL gut regeneration is different from JAJ at this early time point. We interpret this rapid tissue-specific remodeling and recovery of proportional length as an evolutionary pressure to

quickly replace all components of the digestive system in preparation for crucial nutritional input. Surprisingly, at 7 dpm, the ratio of hindgut length to whole-body length is statistically different between JAL and uncut animals. This result might indicate that, after the initial burst to replace the hindgut, the growth of hindgut is slower in JAL animals than in uncut juveniles. It should be noted that the ratio of hindgut length to whole-body length is also significantly different between 7 dpa JAJ and uncut juveniles, indicating similarities between JAL and JAJ gut growth by 7 days into the regeneration process. By 14 dpm, the proportionality of hindgut to whole-body length is recovered, having no significant difference between JAL and uncut juveniles. It is possible that the subtle shift in growth and regeneration rates between JAL and JAJ may be influenced by a minor variation in the initial amputation site between the two groups. In summary, these results indicate that the regeneration of the digestive tract in JAL animals is a dynamic process of remodeling and growth.

The observed remodeling of the hindgut in combination with previous observations of a limited shift in Hox gene expression domains following amputation (de Jong and Seaver, 2016) show that epimorphic and remodeling processes occur concurrently in *C. teleta* regeneration. Similar remodeling has been observed in other annelids (Bely, 2014; Takeo et al., 2008; Zattara, 2020). For example, regionalization of the digestive system in *Enchytraeus japonensis*, as evidenced using histochemistry of alkaline phosphatase and gut-specific markers, re-establishes proportionally following amputation (Myohara et al., 1999; Takeo et al., 2008). In annelids, pre-existing digestive and neural tissues can be remodeled within hours to days after amputation (Bely, 2014; Özpolat and Bely, 2016; Takeo et al., 2008).

Although gut remodeling occurs in *C. teleta*, the cellular source of the new hindgut, either proximal or distal to the cut site, is currently unknown. Our observations of EdU incorporation in the lining of the digestive tract in *C. teleta* is indicative of a local cellular source (Fig. 7). Considering that the embryonic origin of the midgut is from endoderm and the hindgut from ectoderm, it is intriguing that the pre-existing midgut can remodel and exhibit hindgut characteristics. Furthermore, hindgut in tissues proximal to the cut site may have a different cellular source from the regenerating hindgut distal to the cut site. In other annelids, such as *Lumbriculus variegatus*, cells in the gut endoderm may contribute to the regenerated gut (Tweeten and Reiner, 2012). Recent work in *P. dumerilii* reports the presence of gut progenitor cells that contribute to regenerating tissues (Bideau et al., 2024). Outside of annelids, a study in the planarian *Schmidtea mediterranea* indicates that gut regeneration is the combined result of contributions from neoblast stem cells and remodeling of pre-existing tissue (Forsthoefel et al., 2011). Future work is needed to explore whether lineage-restricted stem cell precursors reside in the gut epithelium, or other cellular sources contribute to the regeneration of the digestive system in *C. teleta*.

### Metamorphosis as a key transition event in regenerative potential

Metamorphosis is an event marked by substantial morphological, physiological and ecological changes to develop a new body form (Bishop et al., 2006; Monaghan et al., 2014). These substantive changes can include altered cell plasticity, metabolic differences, stem cell contribution and molecular signaling (e.g. gene transcription, protein synthesis, chromatin accessibility and hormone fluctuations) (Booth et al., 2025). The significant changes associated with metamorphosis may also play a crucial role in changing regenerative potential. Recent ATACseq studies have demonstrated differentially accessible regions of chromatin between pre- and post-

metamorphic stages in flatfish and the hemichordate *Schizocardium californicum* (Bump, 2022; Guerrero-Peña et al., 2023). In anurans, there is evidence that metamorphosis inhibits regeneration-associated gene transcription, either directly or epigenetically (Christen and Slack, 1997; Monaghan et al., 2014; Phipps et al., 2020). Axolotls experimentally induced to undergo metamorphosis experience a reduction in regeneration rate and success (Monaghan et al., 2014), likely due to longer cell cycles and reduced cell proliferation. Hormonal changes can also influence regeneration outcomes. For example, an increase in thyroid hormone directly leads to a loss of regenerative potential in the central nervous system of *X. laevis* (Gibbs et al., 2011). Taken together, our result positioning metamorphosis as a key event in the shift of regeneration potential in *C. teleta* is shared by a range of metazoans.

The molecular, cellular and structural changes associated with metamorphosis serve as a switch in regenerative potential. In some animals, such as anurans, this switch hinders regeneration. However, in some indirect developing invertebrates [e.g. *Antedon bifida* (Vickery et al., 2001), *C. teleta*, etc.], metamorphosis promotes the onset of regenerative success. The relationship between metamorphosis and regeneration may be, in part, due to shared regulatory pathways. Additionally, metamorphosis may result in tissue maturation that renders tissues receptive to signals required for regeneration. Metamorphosis transforms the molecular landscape, including transcriptional and epigenetic elements of an animal, which in turn can affect regenerative ability in either direction.

### Future directions

Despite numerous surveys of annelid juvenile and adult regeneration ability (Bely, 2006; Metzger and Özpolat, 2024; Starunov et al., 2020; Zattara and Bely, 2016), to our knowledge *C. teleta* is the first annelid for which regeneration ability has been systematically characterized across the metamorphic transition. As annelids display a great variation in adult regenerative ability, it can be inferred that this variation extends to earlier developmental stages. Many annelids have larval stages that can be surveyed for regenerative potential. Characterization of additional stages can be conducted on species for which regeneration abilities have already been described as adults and be included in future surveys to gain a better understanding of how regeneration varies with maturation.

### Conclusions

Metamorphosis is a key event in the subsequent acquisition of successful regeneration in *C. teleta*. Our results indicate that posterior regeneration is inherent in post-larval stages. In juveniles, we observed remodeling of pre-existing tissue and epimorphic regeneration of new tissue to replace tissues removed in larvae. Comparisons of regenerative potential in the same animal (e.g. with ontogenetic or physiological boundaries), can provide insights into the impediments and catalysts of regeneration (Goss, 1992). Additional sampling of regenerative ability at different life history stages in other species will provide insight into the onset of regeneration and its relationship with developmental maturation. Varied regeneration abilities across annelids allow us to better understand and parse out the degrees of plasticity that exist across a life cycle.

### Limitations of this study

Owing to a combination of survivorship and small size, not all juveniles were recovered after rearing in mud. Therefore, only two independent experimental replicates were conducted for the re-amputation experiments. In this study a 'replicate' is defined as

an independent experiment consisting of multiple individuals treated at the same time, in the same experimental group. Limited molecular markers were used to characterize regenerating tissues.

## MATERIALS AND METHODS

### Animal care
A colony of *C. teleta* was maintained in glass finger bowls with filtered sea water (FSW) and marine mud at 19°C, as previously described (Grassle and Grassle, 1976). Larvae were dissected from brood tubes of healthy females and raised in FSW until the appropriate age for amputation or induction of metamorphosis. Larvae from a single brood tube were divided into groups of uncut control animals and amputated experimental animals.

### Amputations
Stage 9 larvae were amputated for experiments scored as juveniles and stage 6 larvae were amputated for experiments scored as 3 dpa and 5 dpa larvae. Larval stages were determined using a standard staging system (Seaver et al., 2005). Prior to amputation, larvae were transferred to a methyl cellulose solution (1.6% methyl cellulose in FSW) on a 60 mm Petri dish lid. Once immobilized, larvae were amputated using pulled glass pipettes (Boyd and Seaver, 2023), removing the posterior third of the larva. Subsequently, larvae were transferred to FSW for approximately 5 h to recover and allow wound healing to occur. Larvae representing 'time zero' were then fixed in 4% paraformaldehyde (PFA) in FSW for 1 h at room temperature (for EdU and antibody treatments) or overnight at 4°C (for *in situ* hybridization experiments). Amputated stage 9 larvae were induced to undergo metamorphosis 5 h post-amputation (Boyd and Seaver, 2023). Prior to fixation for the 'time-zero' juveniles, induced animals were visually observed for successful metamorphosis, i.e. burrowing behavior, dropping of ciliary bands, and body elongation. Juvenile worms were fed with marine sediment and underwent water changes weekly until further processing. Animals were sieved from the mud and then placed in plates containing 0.5% cornmeal agar (Sigma-Aldrich) in FSW for 1-4 h to remove attached debris before exposure to EdU or fixation.

JAJ animals were reared in mud in FSW and sifted from mud. In preparation for amputation, individual animals were moved to droplets of a 1:1 solution of 0.37 M MgCl$_2$:FSW on a platform of black dissecting wax (American Educational Products, Fort Collins, CO, USA) for a minimum of 10 min. Worms were amputated using a microsurgery scalpel (FEATHER; 15-degree blade). Using the presence of chaetae and segmental body wall indentations to count individual segments, worms were amputated at the boundary between segments 10 and 11. Following amputation, worms were placed in a 1:1 mixture of 0.37 M MgCl$_2$:FSW for a 1-2 h recovery. Worms were then placed in fresh bowls of FSW with a teaspoon of sieved estuarine mud for 1 week. After the designated time frame, individuals were sifted from the mud, placed into agar plates for debris removal, and then processed for either EdU incorporation or immunohistochemistry.

For the re-amputation experiment, the initial amputation was conducted with stage 9 larvae as described above in this section. The second amputation was performed 14 dpm. At 14 dpm, the cut and uncut juveniles were sifted from the mud and placed in corn-meal agar plates for 1-4 h to remove attached debris. In preparation for amputation, individual animals from either the cut or uncut groups were moved to 1:1 0.37 M MgCl$_2$:FSW droplets on a platform of black dissecting wax (American Educational Products). Worms were amputated with a microsurgery scalpel (FEATHER; 15-degree blade). Using a dissecting microscope (Stemi 2000, Zeiss) to visualize the presence of chaetae and segment indentations, individual segments were counted and worms were amputated at the boundary between segments 10 and 11. Following amputation, worms were placed in a 1:1 solution of 0.37 M MgCl$_2$: FSW for a 1-2 h recovery period and then transferred to a finger bowl containing marine mud and FSW. After 1 week, individuals were sifted from the mud, placed into agar plates for debris removal, and then processed for either EdU incorporation or immunohistochemistry.

### Induction of metamorphosis
Successful metamorphosis is defined and identified by the loss of ciliary bands (prototroch and telotroch), elongation of the body, cessation of swimming and the onset of burrowing behavior. Metamorphosis in *C. teleta* is rapid and reaches completion between 5 and 40 min after induction. Some animals were induced to undergo metamorphosis by exposure to a solution containing vitamin B in FSW as previously described (Boyd and Seaver, 2023). These animals were visually monitored and left in vitamin B solution for up to 1 h to ensure complete metamorphosis prior to fixation as the 'time-zero' group. Any animals that did not undergo metamorphosis were excluded from further analysis. For other experimental groups, intact and amputated animals were placed in glass finger bowls containing filtered seawater and exposed to a thin layer of marine mud to induce metamorphosis. The animals were reared in the same bowls at 20°C until the desired time after metamorphosis. Animals were then removed from the mud and allowed to burrow in 35 mm plastic dishes half filled with 0.5% cornmeal agar (Sigma-Aldrich) in FSW for 1-4 h to remove attached debris.

### EdU incorporation
Juveniles were incubated in a working concentration of 3 µM EdU (Click-iT EdU Alexa Fluor 488 imaging kit, Invitrogen) in FSW for 1 h, with rocking at room temperature. The juveniles were then washed into a 1:1 mixture of 0.37 M MgCl$_2$:FSW for 10 min before being fixed in 4% PFA for 1 h. Following fixation, the specimens were transferred to a glass three-well depression plate, washed into PBS, and incubated in 0.05% Triton X-100 in PBS for 20 min, with rocking at room temperature. They were then incubated with the commercial Click-iT reaction mixture at room temperature for 30 min. The juveniles were washed into PBS and stored at 4°C until antibody labeling.

### Immunohistochemistry
Animals were incubated in a blocking solution of 10% heat-treated normal goat serum in 0.2% Triton X-100 in PBS (PBT) for 45-60 min at room temperature. Anti-acetylated tubulin antibody (goat anti-mouse, T6793, Sigma-Aldrich) was diluted 1:400 in blocking solution and incubated at 4°C overnight, following previously published protocols (Meyer et al., 2015). Following incubation, the animals were washed four times, for 30 min each, in 0.2% PBT. A secondary antibody (goat anti-mouse, Invitrogen) was diluted 1:400 in blocking solution and added to animals for an overnight incubation at 4°C. Animals were once again washed four times, 30 min each, in 0.2% PBT. During the final two washes, Hoechst 33342 (Thermo Fisher Scientific) was added to PBT at a dilution of 1:1000. After the washes, the animals were cleared in 80% glycerol:20% PBS overnight at 4°C prior to imaging. The *vasa* mRNA probe has two expression domains used for scoring: a cluster of cells between segments 5 and 6 and the pgz. The cells in segments 5/6 served as a positive internal control for each specimen; if the cluster was not labeled, that specimen was excluded from further scoring and analysis.

### Hindgut and whole-body length measurements
Animals were imaged on a Zeiss Axio Imager M2 with a red (594) filter set to visualize the anti-acetylated tubulin labeled cilia, with images taken at different focal planes to the digestive system. For hindgut measurements, images were taken to include the entirety of the body. For large specimens, multiple images were taken of a specimen and then digitally stitched together to capture the whole body. Only animals with a clear transition from hindgut to midgut were used for measurements. The curvature of the ciliated hindgut was measured using the segment line tool in ImageJ Fiji. The length of the whole body was similarly measured. A ratio of hindgut length:whole-body length was generated for each specimen using the respective measurements tool. The box and whisker plot of ratio distributions was produced using Statistics Kingdom (https://www.statskingdom.com/advanced-boxplot-maker.html).

### Replicates and statistical analysis
When possible, uncut controls were sourced from the same broods as the cut experimental group (with exception of the 5 dpa larvae). At least two independent replicates, defined as an independent experiment consisting of multiple individuals treated at the same time, in the same experimental group were performed for each time point/treatment. Sample sizes were defined as the total number of specimens scored and included for further analysis for a given experiment.

When necessary, data sets used for statistical analyses were subjected to the Shapiro–Wilk test (https://www.statskingdom.com/shapiro-wilk-test-calculator.html#google_vignette) to determine whether the data fit a normal distribution. The data sets associated with ratio of hindgut length to whole-body length were determined to have a normal distribution and therefore subjected further to paired, two-sample *t*-test (pooled variance) analysis (https://www.statskingdom.com/140MeanT2eq.html). To compare the hindgut:whole body ratio of cut and uncut animals at the time points listed in Table 1, *t*-test statistical analyses (https://www.statskingdom.com/140MeanT2eq.html) were conducted for each cross-group comparison. The data sets for the double-amputation experiment had a normal distribution and were subjected to further *t*-test analyses. An analysis of covariance was conducted on the data set characterizing generation of new segments where days (0-14) were the independent variable, segment number was the dependent variable and cut/uncut was the covariant (MATLAB R2024b Update 5; 24.2.0.2863752).

### Microscopy and imaging

A Zeiss 710 confocal microscope, with 4× bidirectional and unidirectional scan settings for green and red lasers, was used to image the fluorescence of specimens. The resulting files were compiled into *z*-stacks using ImageJ software (Schindelin et al., 2012). A Zeiss Axioskop II motplus compound microscope coupled with a SPOT FLEX digital camera (Diagnostic Instruments Inc.) was used to capture differential interference contrast microscopy images. As noted in figure legends, multiple focal planes of differential interference contrast images were merged using Helicon Focus Software for some images (Helicon Focus 7 & 8). Images were cropped and adjusted (as a whole) for brightness and contrast in either Photoshop (Adobe Photoshop 2020) or Illustrator (Adobe Illustrator 2020). Figures were composed using Adobe Illustrator software (Adobe Illustrator 2020).

### Whole-mount *in situ* hybridization

Animals were collected either immediately following metamorphosis or at 7 dpm, immobilized by incubation in a 1:1 mixture of 0.37 M MgCl$_2$:FSW solution for 10 min, then fixed in 4% PFA in FSW at 4°C overnight. The animals were washed several times with 0.2% Triton X-100 in PBS and then slowly washed into 100% methanol and stored at −20°C. Whole-mount *in situ* hybridization was conducted according to a previously published protocol (de Jong and Seaver, 2017). The digoxigenin-labeled riboprobe was generated using the MEGAscript T7 Transcription kit (Invitrogen) according to the manufacturer's instructions and stored in hybridization buffer (50% formamide, 5× SSC, 50 μg/ml heparin, 0.1% Tween 20, 1% sodium dodecyl sulfate, 50 μg/ml salmon sperm DNA) at −20°C. The *vasa* probe was used at a working concentration of 2 ng/μl and hybridized with animals at 65°C for 48 h. The animals were incubated with an anti-digoxigenin-alkaline phosphatase conjugated antibody (Roche), such that the RNA probe can be visualized upon exposure to a color reaction solution with NBT/BCIP (nitro blue tetrazolium chloride/5-bromo-4-chloro-3-indolyphosphate; Promega). The animals were visually monitored during development, and all treatment groups were stopped at the same time as controls. Color development was stopped with extensive washes of 0.2% Triton X-100 in PBS and a final fixation in 4% PFA in FSW for 30 min. Tissues were cleared in 80% glycerol in 1× PBS before being mounted on microscope slides for analysis and imaging.

### Determination of segment number

The number of segments in a juvenile was determined by scoring three independent markers: ganglia number using the Hoechst nuclear stain, paired peripheral nerves using anti-acetylated tubulin antibody labeling, or chaetae (biofluorescent in multiple channels). If two out of three markers scored, i.e. ganglia in the nuclear stain, paired peripheral nerves in antibody staining, or chaetae, yielded different results, then the common number was ruled the final number of total segments (frequently, chaetae were fewer in number compared to the presence of ganglia and peripheral nerves since chaetal development occurs very late in segment formation). On the rare occasion when all three markers yielded different segment numbers (e.g. 14,15,16), the following procedure was followed: each channel was scrutinized again and recounted. If still in disagreement, then an average of the three numbers was assigned as the final number of segments for that animal.

Changes in the appearance of the VNC were used to determine the boundary between pre-existing and regenerated segments. The connectives of the VNC are severed during amputation, such that there is a substantial decrease in thickness of these nerves as they cross the amputation plane. Tissue distal to the position of this transition of the VNC were identified as newly regenerated. The distinct reduction in body width distal to the amputation plane was also indicative of new growth.

### Acknowledgements
We thank Dr Danielle de Jong for cloning the *C. teleta vasa* gene and generating the RNA probe used in this study, and Brent Foster for critical reading of the manuscript. We are grateful to Dr Cezar Borba for help with the statistical analyses.

### Competing interests
The authors declare no competing or financial interests.

### Author contributions
Conceptualization: A.A.B., E.C.S.; Data curation: A.A.B.; Formal analysis: A.A.B., E.C.S.; Funding acquisition: E.C.S.; Investigation: A.A.B., E.C.S.; Methodology: A.A.B.; Project administration: E.C.S.; Resources: E.C.S.; Supervision: E.C.S.; Validation: A.A.B., E.C.S.; Visualization: A.A.B., E.C.S.; Writing – original draft: A.A.B., E.C.S.; Writing – review & editing: A.A.B., E.C.S.

### Funding
This work was funded by the National Science Foundation (IOS 2316882 to E.C.S.). Open Access funding provided by the University of Florida. Deposited in PMC for immediate release.

### Data and resource availability
All relevant data and details of resources can be found within the article and its supplementary information.

### The people behind the papers
This article has an associated 'The people behind the papers' interview with some of the authors.

### Peer review history
The peer review history is available online at https://journals.biologists.com/dev/lookup/doi/10.1242/dev.204995.reviewer-comments.pdf

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
