## [Peer Review File · Development (Cambridge, England)]

Regrowing the growth zone: metamorphosis kickstarts regeneration in the annelid *Capitella teleta*

Alicia A. Boyd and Elaine C. Seaver

DOI: 10.1242/dev.204995

Editor: Mansi Srivastava

Review timeline

Original submission:	30 May 2025
Editorial decision:	28 July 2025
First revision received:	30 October 2025
Editorial decision:	1 December 2025
First revision received:	18 December 2025
Accepted:	20 December 2025

Original submission

First decision letter

MS ID#: dev.204995

MS TITLE: Regrowing the growth zone: metamorphosis kickstarts regeneration in the annelid *Capitella teleta*

AUTHORS: Alicia A. Boyd and Elaine C. Seaver

Dear Dr Seaver,

I have now received all the referees' reports on the above manuscript, and have reached a decision. The referees' comments are appended below, or you can access them online: please go to:

As you will see, the referees find your work to be of interest, but have some significant criticisms and recommend a revision of your manuscript before we can consider publication. In particular, reviewers would like to see statistical analyses, better approaches for quantification (e.g. for the gut), and also clearer articulation of definitions (e.g. of 'morphallaxis') and hypotheses. If you are able to revise the manuscript along the lines suggested, which may involve further experiments, I will be happy receive a revised version of the manuscript.

Your revised paper will be re-reviewed by one or more of the original referees, and acceptance of your manuscript will depend on your addressing satisfactorily the reviewers' major concerns. Please also note that Development will normally permit only one round of major revision. If it would be helpful, you are welcome to contact us to discuss your revision in greater detail. Please send us a point-by-point response indicating your plans for addressing the referees' comments, and we will look over this and provide further guidance.

Please attend to all of the reviewers' comments and ensure that you clearly highlight all changes made in the revised manuscript. Please avoid using 'Tracked changes' in Word files as these are lost in PDF conversion. I should be grateful if you would also provide a point-by-point response detailing how you have dealt with the points raised by the reviewers in the 'Response to Reviewers' box. If you do not agree with any of their criticisms or suggestions please explain clearly why this is so.

Reviewer 1*Advance summary and potential significance to field*

The manuscript by Boyd and Seaver described the onset of regeneration competency following metamorphosis in the annelid worm *Capitella teleta*. The relationship between regeneration and metamorphosis is very interesting especially in the light of how variable it can be in different animal representatives. In most animals wound healing and regeneration abilities appear to decrease after development and with the onset of maturation and ageing as seen in mammals, and even in some amphibians where regeneration competency is lost or reduced following metamorphosis. A few scarce examples surprisingly gain regeneration competency following metamorphosis, including the annelid presented here. Therefore, understanding the biological processes which underlie plasticity of regeneration competencies can be very well studied using comparative approaches in the same animals at different life stages.

The authors neatly designed a set of experiments in which the larvae are posteriorly amputated, induced to metamorphose and then compared to both uncut juvenile worms as well as juxtaposed animals injured as larvae compared to injured as juveniles, and in a final experiment compared single-amputated animals (AL) compared to double-amputated (as larvae and then again as juvenile). Using a comprehensive set of parameters including segment number, body length, length of gut, reformation of the posterior growth zone (vasa expression) and cell proliferation (EdU) the authors demonstrate that the animals which were amputated to remove 1/3 of the posterior end of the worm can reform the growth zone after metamorphosis and then proceed to add segments comparably with uncut juveniles. Analysing the regeneration of the missing hindgut the authors show evidence for morphallaxis together with cell proliferation as putative mechanisms of hindgut restoration. The restoration of the growth zone following metamorphosis and the gain of regeneration competency provides several fascinating questions to explore in future work: what is the mechanism by which juvenile animals have the capability to restore the pgz which is lacking in the larva? What is the relationship between nutritional status, metamorphosis and regeneration competence? What are the molecular differences between the pgz of an uncut larva, juvenile and after regeneration?

Altogether this manuscript is thoroughly written and provides an interesting addition to the field. As a limitation, the study is largely observational and thus not many mechanistic insights can be drawn, though it provides a solid foundation for future work.

Comments for the author

I have several comments that the authors should address in their revised version.

Figures:

- Please put the n=X labels at the bottom left of each figure and add staining info in the top right corner; for example - Hoechst for nuclei, Vasa for ISH, antibody name/EdU in the same colour as represented in image
- In figure 1: the schematics of the experiments are a bit confusing. Why not have one single schematic showing the whole timeline of experiments, namely:
 - o Larva (which stage?) - cut (when?), metamorphosis induction (when?) timepoint zero (fixation I), timepoint 3dpm (fixation II) timepoint 7dpm (fixation III) and timepoint 14dpm (fixation IV)
 - o I think that could just be one long panel instead of two separate ones.
- Figure 2: legend: which nuclear stain?
- If the same anti-acetylated tubulin antibody was used for both the ventral nerve cord images and the hindgut images, why do they look so different when comparing images from Figure 2 and images from figure 5? Is a different focal plane imaged? Please clarify
- In the 14 days confocal images there is no amputation plane indicated, is it too far off to the anterior and thus not visible or is it not marked?

It would be great to have a summary table of segment counts for each experiment.

Methods:

- The section on amputations is somewhat confusing. Why are stage 6 larvae used for ISH, where it then says at 3 and 5 days pa, whereby in the figure it states 7 days cut and uncut (same in methods section on ISH)? Why is stage 9 used for the remaining regeneration experiments?
- Methods about amputation and metamorphosis induction states "as previously described" but no reference is provided
- How soon after the amputation was metamorphosis induced? How long does metamorphosis take?
- How is the amputation plane identified in 7 and 14 days post metamorphosis animals?
- The method section describing the double-cut experiment is missing. In the limitations section the authors mention that there was a low recovery of re-amputation animals - what does that mean? Did they not survive the second amputation or did they not regenerate fully? What are the two independent biological replicates (2 batches or larvae? There are 27 counts for the experiment mentioned in the results section) Please clarify

Reviewer 2

Advance summary and potential significance to field

In this article, Boyd and Seaver follow up on previous findings that posterior regeneration ability, widespread across the Annelida, seems to be limited or absent in late larval stages of *Capitella teleta* (Sedentaria: Capitellidae). In this work, they show that this limitation is removed after metamorphosis from larval to juvenile stage. The author performed a series of experiments showing that after inducing metamorphosis, individuals that had their posterior end amputated as larva (and thus missing such posterior end) will reconstruct it to a point in which is almost indistinguishable from the posterior end of a juvenile that has never been amputated as a larva. Metamorphosed amputated larvae made new segments and a new posterior growth zone, as evidenced by segmental structures and pgz-specific patterns of cell proliferation and marker gene expression. Interestingly, they showed that morphallactic events - namely, the re-specification of midgut lining tissue to hindgut lining tissue - could occur in tissues proximal to the amputation site, but not within the regenerated region, even in the non-regenerative larval context. Last, but not least, they proved that having experienced the loss of posterior end (either as larva or as a juvenile) and subsequent regeneration as juveniles did not hinder the ability to regenerate after new amputations. This study is interesting and provides compelling evidence that, at least in some groups, regeneration ability can increase as life history progresses. It is thus a great counterexample to the general understanding that regenerative abilities are always highest at the earliest developmental stages and become progressively restricted during post-embryonic life. All experiments are well planned and adequately executed, resulting in clean and clear-cut results. I have a number of minor comments spread throughout the manuscript that have been made in a PDF document, and most of them relate to suggestions for data display. One point I think is worth considering is that the authors interpret their results as meaning that the molecular signals that usually boot regeneration upon injury and tissue loss linger on after regenerative failure in cut larvae and through metamorphosis, so that when the worm becomes regeneration-competent, regeneration is resumed and completed. However, there is an alternate interpretation: that upon metamorphosis, sensing that structures are missing could trigger regeneration independently of injury. This is an interesting alternative hypothesis, as it implies that the developmental regulatory networks that drive regeneration can be rewired to be set in motion by triggers other than injury - as seen during asexual reproduction by fission.

There are a couple places where a proper statistical analysis would be indicated: in lines 159-161, the number of segments generated since metamorphosis is compared, but not statistically tested. Given the sample size, even a traditional t-test could be used (although it is understandable if there is an institutional preference for the Mann-Whitney test). Same is true for results of re-amputation experiments (lines 285-288): with sample sizes of 44 and 27, there is enough data for proper testing, and also for a more explicit display of the distribution of added segment counts. Here I was confused by the disclaimer in the final section about "only two independent biological replicates". The term "twice cut" appears only then, so I am not sure what it is referring to. Something similar is to be said about the analysis of midgut/hindgut re-specification. Description of results and images are quite convincing on their own, so the actual conclusion is held well. However, taking linear measurements of a curvy structure like the gut could introduce considerable noise, especially for an organ whose functions depend on relative surface area ratios that depend a

lot on the actual length of the organ itself rather than how it is positioned along the body. The authors even acknowledge that this could be a source of noise. Since from the methods it seems like they imaged the animals and then measured using software, it should be straightforward (though obviously some extra labor would be involved) to re-measure midgut and hindgut as they themselves suggest and re-run the statistical analyses. While at it, please check if multiple MW U tests without correcting for experiment-wise error is the right way to analyze the data, or else a non-parametric ANOVA followed by a pairwise tests but applying a multiple comparison correction (like Bonferroni, Holm, or Benjamini-Hochberg) to the p-value.

None of the above comments are critical regarding the main take-home messages of this article. Once published, this paper will make a significant contribution to the annelid regeneration field, and prompt very interesting follow-ups, in this species and in other systems.

Eduardo Zattara

The reviewer also provided comments directly on the manuscript file, which are not included in this report for technical reasons.

Reviewer 3

Advance summary and potential significance to field

Boyd and Seaver address the role of metamorphosis in onset of a constitutive posterior growth program, and posterior regeneration competence, in the annelid *Capitela teleta*. Juvenile animals previously amputated as larvae acquire competence to regenerate following induction of metamorphosis. The authors nicely introduce the model system and its anatomy and provide ample background and discussion of the significance of the question of when and how regenerative abilities are gained or lost across the life cycle. This question is important and will be of interest to a broad swath of the Development readership. The authors clearly describe experiments aimed at demonstrating that *Capitela teleta* gains posterior regenerative abilities post-metamorphosis. The mechanism eludes us presently, but the authors describe several intriguing and testable hypotheses.

Comments for the author

My concerns with the manuscript in its present form stem from what are likely technical limitations with existing WISH and anti-acetylated tubulin immunostaining protocols, and interpretation of the data on gut morphallaxis (explicit definition of the term morphallaxis is needed; the use of a single stain to score hindgut identity is also troubling). The authors are aware of some of these issues (addressed in their limitations), and further clarification and support of key points will be necessary for publication of the work. I appreciate that there are a number of interesting lines of inquiry to pursue in this system and anticipate that optimization of the staining protocols will help you to vet these hypotheses. Finally, there are a few areas where the authors can substantiate claims of sameness or distinction by applying statistical tests to data they have already acquired.

Recommended protocol optimization tip, related to colorimetric WISH in Figure 4: I recommend looking into whether developing with Fast Blue or Vector Blue and visualizing the product fluorescently in the far-red channel may improve signal intensity, sensitivity, and cell-level resolution relative to what you show here with NBT-BCIP. You can try altering the development steps, using Fast Blue instead of NBT-BCIP, while keeping earlier stages of the WISH workflow intact (though in our experience, optimizing sample penetration using Proteinase K digestion and post-fixation prior to the prehybe steps has greatly helped, with both NBT-BCIP and the Fast Blue substrates). See protocols 10.1186/s13227-016-0044-8 and 10.1242/dev.204775.

Major concerns that should be addressed prior to publication:

Anti-acetylated tubulin staining: Do you have supporting evidence that the lack of anti-acetylated tubulin staining in the midgut is not a technical artifact due to problems with tissue permeabilization? The foregut staining is weak and difficult to detect in the 3D cut and uncut juveniles - is this due to the planes imaged/shown, or to suboptimal penetration? Is there ultrastructure-based data or other evidence that would confirm the midgut is not ciliated, or WISH/FISH data using with riboprobes that would specifically mark ciliated cells within the gut or

other compartment-specific markers? Lack of corroborating evidence with additional markers and the possibility of technical artefacts influencing your conclusions is troubling. Fixation with TCA, or a combination of TCA and PFA, may also improve staining with this antibody. You should reference the specific antibody you are using in the methods since several antibodies exist.

Lines 204-205: Have you quantified ciliated hindgut length to substantiate this conclusion?

Linearity of gut measurements - you raise caveats associated with linear marking of the ends of the gut territory to make these measurements. Would segmenting and scoring gut surface areas on the 2D images be a more accurate representation of the amount of hindgut present?

Line 241-247: Why were different time points evaluated for JAL and JAJ amputations? Why was full regeneration of the gut not scored for the JAJ amputations at 14 dpc? Are different amounts of tissue removed for the JAL and JAJ amputations, and may this factor into differences in the rates of recovery?

You have not defined the term morphallaxis for readers, though my understanding of the classical usage of the term is that it refers to regenerative remodeling that does not require cell division. I question whether applying this term is appropriate given that you show local sources of EdU+ incorporation into the gut. While you may have a change in state or fate of the gut cells during regeneration, your data are not consistent with remodeling in the absence of cell division. You may have ways of testing this, but you don't support that claim here. I would suggest tempering the language to more accurately reflect what you are studying: regeneration occurring contemporaneously with a constitutive posterior growth program, that involves production of new cells and potentially tissue remodeling. I think these phenomena are interesting, but it is difficult to tease apart what changes are truly cell proliferation-independent without introducing paradigms that halt proliferation (and potentially induction of proliferative pgz+ cells).

Related to the previous point: You clearly show incorporation of EdU+ signal in short (1 hour) pulses within the gut at 3 and 14 day post-cut (Figure 6, cut juveniles, but do not show the same for the amputated larvae at 5 dpa. It's clear that there is ciliated hindgut tissue that stains with anti-acetylated tubulin - but I don't think that you can exclude the possibility that this occurs in the absence of cell division (you don't reference any studies that would suggest there is a global halt to cell division during this stage/condition, and you would clearly have the protocols in place to test for EdU incorporation within the new hindgut in these larval amputations). Without further experimental evidence you cannot conclude that hindgut appearance in larvae at 5 dpa is "morphallaxis" (proliferation-independent).

Minor points for revision or discussion (future directions, or consider citing prior studies that would add to the current text descriptions):

Lines 156-161: The variability in the number of new segments produced between 3-7 days post-metamorphosis in but cut and uncut animals is interesting. Are these new segments formed sequentially or contemporaneously? Is anything known about what influences this variation? With the data that you have on hand, it would be possible to compare the medians of your cut vs uncut samples to determine whether the medians are significantly different or not (e.g., with a Mann-Whitney U test).

Is it known whether the vasa and piwi are induced within differentiated cells, that then re-enter the cell cycle to re-form the pgz? The identity of the cells that first express GMP genes, whether this is a homo- or heterogeneous population (e.g., are there ectodermal and mesodermal sources that contribute to the pgz, and are these equivalent in their potential) would be interesting avenues for future study.

White arrowheads and the amputation plane in the Figure 2 images (nuclei and nervous system panel) are difficult to distinguish from the images and could be bolder and colored so that they stand out.

Line 280: 7 dpm - should this read 7 dpa (days post-amputation)?

Your discussion point that onset of regeneration competence may be linked to nutritional sources is interesting, and there is some work in *Xenopus* suggesting that the "refractory" period can be overridden by modulating nutrient stores 10.1016/j.ydbio.2021.01.005.

First revision

Author response to reviewers' comments

Note: responses to detailed reviewer comments are in bold.

Reviewer 1:

I have several comments that the authors should address in their revised version.

Figures:

- Please put the n=X labels at the bottom left of each figure and add staining info in the top right corner; for example - Hoechst for nuclei, Vasa for ISH, antibody name/EdU in the same colour as represented in image

We added the staining information to the figures in either the bottom left corner of the panel or above the images in the same color as used in the images. We also moved the text information regarding staining to the top of the figure legends.

- In figure 1: the schematics of the experiments are a bit confusing. Why not have one single schematic showing the whole timeline of experiments, namely:

o Larva (which stage?) - cut (when?), metamorphosis induction (when?) timepoint zero (fixation I), timepoint 3dpm (fixation II) timepoint 7dpm (fixation III) and timepoint 14dpm (fixation IV)
o I think that could just be one long panel instead of two separate ones.

We thank the reviewer for this suggestion and have rearranged the schematic in figure 1 to be one long panel as recommended. We also added the larval stage, wound healing, and renumbered the fixations to include the time zero fixation.

- Figure 2: legend: which nuclear stain?

We updated the Figure 2 legend to read 'Nuclei are stained with Hoechst.'

- If the same anti-acetylated tubulin antibody was used for both the ventral nerve cord images and the hindgut images, why do they look so different when comparing images from Figure 2 and images from figure 5? Is a different focal plane imaged? Please clarify

The ventral nerve cord and the hindgut are in distinctly different focal planes when viewed in ventral view such as are the images presented in Figs 2 and 5. We added the following clarification statement to the "Hindgut and whole-body length measurements" methods section: '...with images taken at different focal planes to visualize either the ventral nerve cord or the digestive system.'

- In the 14 days confocal images there is no amputation plane indicated, is it too far off to the anterior and thus not visible or is it not marked?

The following statement was added to the Figure 2 legend: '(V, X) In 14 day images, the amputation site is anterior to the section imaged and out of the field of view. The amputation site is therefore not marked in these panels.'

It would be great to have a summary table of segment counts for each experiment.

A summary table of the new segments formed and associated statistics was generated and added to the revised version of the manuscript (Table 1).

Methods:

- The section on amputations is somewhat confusing. Why are stage 6 larvae used for ISH, where it then says at 3 and 5 days pa, whereby in the figure it states 7 days cut and uncut (same in methods section on ISH)? Why is stage 9 used for the remaining regeneration experiments?
The ‘Amputations’ section in the methods has been rewritten for clarity.

- Methods about amputation and metamorphosis induction states "as previously described" but no reference is provided
The reference Boyd and Seaver, 2023 was added after this phrase.

- How soon after the amputation was metamorphosis induced? How long does metamorphosis take?
The following statement was added to the ‘Amputations’ section of the methods: “Amputated stage 9 larvae were induced to undergo metamorphosis 5 hr post-amputation by either exposure to vitamin B in FSW (time zero animals) or transferred to a glass finger bowl with marine mud and FSW (Boyd and Seaver, 2023).”
Metamorphosis is a rapid event in *C. teleta*. The following information was added to the methods section “Induction of metamorphosis”: “Successful metamorphosis is defined and identified by the loss of ciliary bands (prototroch and telotroch), elongation of the body, cessation of swimming and the onset of burrowing behavior. Metamorphosis in *C. teleta* is rapid and reaches completion between 5-40 minutes after induction. Some animals were induced to undergo metamorphosis by exposure to a solution containing vitamin B in FSW as previously described (Boyd and Seaver, 2023). These animals were visually monitored and left in vitamin B solution for up to one hour to ensure complete metamorphosis prior to fixation as the ‘time zero’ group. For other experimental groups, intact and amputated animals were placed in glass finger bowls containing filtered seawater and exposed to a thin layer of marine mud to induce metamorphosis.’

- How is the amputation plane identified in 7 and 14 days post metamorphosis animals?
To explain how the location of the amputation plane was determined, the following statements are included at the end of the ‘Determination of segment number’ section of the Methods: ‘Changes in the appearance of the ventral nerve cord (VNC) were used to determine the boundary between pre-existing and regenerated segments. The connectives of the ventral nerve cord are severed during amputation, such that there is a substantial decrease in thickness of these nerves as they cross the amputation plane. Tissue distal to the position of this transition of the VNC were identified as newly regenerated. The distinct reduction in body width distal to the amputation plane was also indicative of new growth.’

- The method section describing the double-cut experiment is missing.
We thank the reviewer for catching this inadvertent omission. We added the following paragraph to the end of the ‘Amputations’ section of the Methods:

“For the re-amputation experiment, the initial amputation was conducted with stage 9 larvae as described above in this section. The second amputation was performed 14 dpm. At 14 dpm, the cut and uncut juveniles were sifted from the mud and placed in cornmeal agar plates for 1-4 hours to remove attached debris. In preparation for amputation, individual animals from either the cut or uncut groups were moved to 1:1 0.37M MgCl₂: FSW droplets on a platform of black dissecting wax (American Educational Products, Fort Collins, CO, USA). Worms were amputated with a microsurgery scalpel (Feather; 15-degree blade, Carlsbad, CA, USA). Using the presence of chaetae and segment indentation to count individual segments, worms were amputated at the boundary between segments 10 and 11. Following amputation, worms were placed in 1:1 0.37M MgCl₂: FSW solution for a 1-2 hour recovery period and then transferred to a finger bowl containing marine mud and FSW. After 1 week, individuals were sifted from the mud, placed into agar plates for debris removal, and then processed for either EdU incorporation or immunohistochemistry.”

In the limitations section the authors mention that there was a low recovery of re-amputation animals - what does that mean? Did they not survive the second amputation or did they not regenerate fully? What are the two independent biological replicates (2 batches or larvae? There are 27 counts for the experiment mentioned in the results section) Please clarify

In the methods section ‘Replicates and statistical analysis’, we added the following sentence: “At least 2 independent replicates, defined as an independent experiment consisting of multiple individuals treated at the same time, in the same experimental group were performed for each timepoint/treatment.”

Reviewer 2:

I have a number of minor comments spread throughout the manuscript that have been made in a PDF document, and most of them relate to suggestions for data display.

We thank the reviewer for recommending these minor comments. We have thoughtfully incorporated a response to almost all these comments into the revised manuscript.

One point I think is worth considering is that the authors interpret their results as meaning that the molecular signals that usually boot regeneration upon injury and tissue loss linger on after regenerative failure in cut larvae and through metamorphosis, so that when the worm becomes regeneration-competent, regeneration is resumed and completed.

However, there is an alternate interpretation: that upon metamorphosis, sensing that structures are missing could trigger regeneration independently of injury. This is an interesting alternative hypothesis, as it implies that the developmental regulatory networks that drive regeneration can be rewired to be set in motion by triggers other than injury - as seen during asexual reproduction by fission.

We thank the reviewer for raising this alternative hypothesis. We have added it to the discussion section (lines 315-317) and to the future directions section.

There are a couple places where a proper statistical analysis would be indicated: in lines 159-161, the number of segments generated since metamorphosis is compared, but not statistically tested. Given the sample size, even a traditional t-test could be used (although it is understandable if there is an institutional preference for the Mann-Whitney test). Same is true for results of re-amputation experiments (lines 285-288): with sample sizes of 44 and 27, there is enough data for proper testing, and also for a more explicit display of the distribution of added segment counts. **Statistical analysis was performed to compare how the uncut and cut worms generated new segments over the time points sampled. For this analysis, we used an analysis of covariance test (ANACOVA) of the number of total segments over the 14 day period in uncut and cut animals. More explicit details on the distribution of recorded segments were added to the text as well as presented in a table (Table1).**

Here I was confused by the disclaimer in the final section about "only two independent biological replicates". The term "twice cut" appears only then, so I am not sure what it is referring to.

In the methods section 'Replicates and statistical analysis', we added the following sentence: "At least 2 independent replicates, defined as an independent experiment consisting of multiple individuals treated at the same time, in the same experimental group were performed for each timepoint/treatment." 'Twice cut' was changed to 'double cut' in the text

Something similar is to be said about the analysis of midgut/hindgut re-specification. Description of results and images are quite convincing on their own, so the actual conclusion is held well. However, taking linear measurements of a curvy structure like the gut could introduce considerable noise, especially for an organ whose functions depend on relative surface area ratios that depend a lot on the actual length of the organ itself rather than how it is positioned along the body. The authors even acknowledge that this could be a source of noise. Since from the methods it seems like they imaged the animals and then measured using software, it should be straightforward (though obviously some extra labor would be involved) to re-measure midgut and hindgut as they themselves suggest and re-run the statistical analyses.

We followed the reviewer's recommendation and remeasured the hindgut tissue following its coiled shape for all samples using Fiji's segmented line and measurement tools. We also conducted statistical analysis using the new measurements data. This results section was rewritten to include these more accurate measurements (see first paragraph of 'Remodeling of preexisting tissue in digestive track').

While at it, please check if multiple MW U tests without correcting for experiment-wise error is the right way to analyze the data, or else a non-parametric ANOVA followed by a pairwise tests but applying a multiple comparison correction (like Bonferroni, Holm, or Benjamini-Hochberg) to the p-value.

After we remeasured the lengths of the hindguts to account for their coiled nature, our data sets became normally distributed. Since they were normally distributed across all time points, we were able to conduct t-test analyses.

Reviewer 3:

SUGGESTIONS TO AUTHORS

My concerns with the manuscript in its present form stem from what are likely technical limitations with existing WISH and anti-acetylated tubulin immunostaining protocols, and interpretation of the data on gut morphallaxis (explicit definition of the term morphallaxis is needed; the use of a single stain to score hindgut identity is also troubling). The authors are aware of some of these issues (addressed in their limitations), and further clarification and support of key points will be necessary for publication of the work. I appreciate that there are a number of interesting lines of inquiry to pursue in this system and anticipate that optimization of the staining protocols will help you to vet these hypotheses. Finally, there are a few areas where the authors can substantiate claims of sameness or distinction by applying statistical tests to data they have already acquired.

Recommended protocol optimization tip, related to colorimetric WISH in Figure 4: I recommend looking into whether developing with Fast Blue or Vector Blue and visualizing the product fluorescently in the far-red channel may improve signal intensity, sensitivity, and cell-level resolution relative to what you show here with NBT-BCIP. You can try altering the development steps, using Fast Blue instead of NBT-BCIP, while keeping earlier stages of the WISH workflow intact (though in our experience, optimizing sample penetration using Proteinase K digestion and post-fixation prior to the prehybe steps has greatly helped, with both NBT-BCIP and the Fast Blue substrates). See protocols 10.1186/s13227-016-0044-8 and 10.1242/dev.204775.

We thank the reviewers for the technical tips. An established protocol for colorimetric WISH is available for our study animal *Capitella teleta*, and this protocol has been successfully used to characterize the expression of over 100 genes at different stages of the life cycle, many of which are published. Examples include genes expressed across a range of different cell types including those positioned in deep tissue and in single cells (Neal et al., 2019; de Jong and Seaver, 2018; Sur et al., 2017; de Jong, and Seaver, 2016; Boyle et al., 2014; Boyle and Seaver, 2008; Seaver and Kaneshige, 2006). In Figure 4, our goal was simply to show the induction of expression of a stem cell marker by amputation. We repeated the *vasa* WISH experiment, obtained reproducible results and increased our sample size. We were not trying to obtain cell-level resolution in this experiment.

Major concerns that should be addressed prior to publication:

Anti-acetylated tubulin staining: Do you have supporting evidence that the lack of anti-acetylated tubulin staining in the midgut is not a technical artifact due to problems with tissue permeabilization? The foregut staining is weak and difficult to detect in the 3D cut and uncut juveniles - is this due to the planes imaged/shown, or to suboptimal penetration? Is there ultrastructure-based data or other evidence that would confirm the midgut is not ciliated, or WISH/FISH data using with riboprobes that would specifically mark ciliated cells within the gut or other compartment-specific markers? Lack of corroborating evidence with additional markers and the possibility of technical artefacts influencing your conclusions is troubling. Fixation with TCA, or a combination of TCA and PFA, may also improve staining with this antibody. You should reference the specific antibody you are using in the methods since several antibodies exist.

The anti-acetylated tubulin antibody is a commonly used reagent in our laboratory and we have previously used this as a marker of the nervous system and ciliated structures in *C. teleta* (e. g. Meyer et al., 2015; de Jong and Seaver, 2016; Lanza and Seaver, 2020; Boyd and Seaver, 2023), and specifically used this reagent to distinguish the midgut from the hindgut (Kunselman and Seaver, 2025). We inadvertently omitted specifics here so we thank the reviewer for bringing attention to these details. Additional File 6a in Meyer et al., 2015 shows that the anti-acetylated tubulin antibody clearly distinguishes the midgut from the hindgut. In the midgut, the elaborate web of neuronal processes of the enteric nervous system are visible, whereas in the hindgut, the prominent labeling is the cilia in the lumen of the gut. Therefore, this antibody provides a clear distinction between these two subregions of the alimentary canal.

Both structures recognized by the anti-acetylated antibody are in similarly deep tissues. Thus, we do not think the labeling we observe is the result of poor permeability.

Because of its cross reactivity, this antibody reagent is widely used across invertebrates, including in other annelids and mollusks. With respect to riboprobes that are expressed by ciliated cells, the spiralian-specific genes *trochin* and *lophotrochin* are expressed in all ciliary bands of the larva, but in juveniles stages, only *trochin* is expressed and it is limited to the ventral face of the pharynx and thus, these are not useful reagents for our study (Wu et al., 2020).

We added a higher magnification image to Figure 5 (panel D') to show the clear transition and distinct labeling patterns between midgut and hindgut when visualized with the anti-acetylated tubulin antibody.

We provide the details of the antibody, including the catalogue number for the exact anti-acetylated tubulin antibody to the 'Immunohistochemistry' methods section.

We also modified the text of the results with more accurate wording, which now reads: "The digestive system of *C. teleta* is regionalized and broadly divided into a foregut, midgut, and hindgut (Fig. 5A - D'). These regions can be distinguished by differences in labeling patterns with an anti-acetylated tubulin antibody as previously reported (Additional File 6a, Meyer et al., 2015). In the foregut and hindgut, the most prominent labeling includes cilia in the digestive track lumen, whereas in the midgut, a complex web of neurites of the enteric nervous system are the most visible feature (Fig. 5A, B, D'; n = 47/47). Amputations in the midgut provides the advantage of being able to distinguish between midgut and hindgut identity in regenerating tissues."

The pictures we included in the manuscript were zstacks produced from multiple images to show the entirety of the hindgut region. Unfortunately, in doing so, the finer cilia often become blurred. We have also included a picture (below) to highlight the fine features of the cilia in the hindgut that allows us to confidently distinguish it from other antibody labeling.

NOTE: We have removed unpublished data that had been provided for the referees in confidence.

Lines 204-205: Have you quantified ciliated hindgut length to substantiate this conclusion?

We remeasured the hindgut tissue following its coiled shape for all samples using Fiji's segmented line and measurement tools following this and another reviewer's recommendation. We also conducted statistical analysis using the new measurements data. The results section was rewritten to include these more accurate measurements (see first paragraph of 'Remodeling of preexisting tissue in digestive track').

Linearity of gut measurements - you raise caveats associated with linear marking of the ends of the gut territory to make these measurements. Would segmenting and scoring gut surface areas on the 2D images be a more accurate representation of the amount of hindgut present?

We followed this and another reviewer's recommendation and remeasured the hindgut tissue following its coiled shape for all samples using Fiji's segmented line and measurement tools. We also conducted statistical analysis using the new measurements data. This results section was rewritten to include these more accurate measurements (see first paragraph of 'Remodeling of preexisting tissue in digestive track').

Line 241-247: Why were different time points evaluated for JAL and JAJ amputations? Why was full regeneration of the gut not scored for the JAJ amputations at 14 dpc? Are different amounts of tissue removed for the JAL and JAJ amputations, and may this factor into differences in the rates of recovery?

Our experimental design was intended to give JAL maximum time to regenerate as we knew they regenerated slower than larvae; therefore, juveniles were examined at 7 dpa. Previous published work has shown that in juvenile posterior regeneration, all tissues are replaced by

7dpa (de Jong and Seaver, 2016). In addition, it was not our intention to track gut remodeling long term, and since our data show that the difference in ciliated hindgut lengths between treatments is insignificant by 7dpa/5dpa, we feel it not necessary extend this experiment to a later time point .

It is possible that slightly different amounts of tissues are removed between JAL and JAJ experiments, and this may factor into differences in recovery rate. Although we can achieve a high level of precision for amputations in juveniles (by counting segments), the very small size of segments in larvae precludes the same level of precision. However, amputations are in the same region of the body in both groups are only about one segment different on average (juveniles amputated at segment 10 and cut site in larva resulting in an average of 9.6 segments remaining (see Table 1) and into the same subregion of the gut, i.e. midgut, fully removing the hindgut. The location of amputations was confirmed in time zero amputees. In the discussion section '*Capitella teleta* replaces lost tissues only after metamorphosis' we added the following sentence: "We acknowledge that the subtle shift in growth and regeneration rates between JAL and JAJ may be influenced by minor variation in the initial amputation site between the two groups."

You have not defined the term morphallaxis for readers, though my understanding of the classical usage of the term is that it refers to regenerative remodeling that does not require cell division. I question whether applying this term is appropriate given that you show local sources of EdU+ incorporation into the gut. While you may have a change in state or fate of the gut cells during regeneration, your data are not consistent with remodeling in the absence of cell division. You may have ways of testing this, but you don't support that claim here. I would suggest tempering the language to more accurately reflect what you are studying: regeneration occurring contemporaneously with a constitutive posterior growth program, that involves production of new cells and potentially tissue remodeling. I think these phenomena are interesting, but it is difficult to tease apart what changes are truly cell proliferation-independent without introducing paradigms that halt proliferation (and potentially induction of proliferative pgz+ cells).

We agree that the textbook term morphallaxis refers to tissue remodeling in the absence of cell division as exhibited during *Hydra* regeneration. Here, we did not mean imply a mechanism of how the gut tissue changes identity, so we replaced morphallaxis with the term 'tissue remodeling' throughout the text.

Related to the previous point: You clearly show incorporation of EdU+ signal in short (1 hour) pulses within the gut at 3 and 14 day post-cut (Figure 6, cut juveniles, but do not show the same for the amputated larvae at 5 dpa. It's clear that there is ciliated hindgut tissue that stains with anti-acetylated tubulin - but I don't think that you can exclude the possibility that this occurs in the absence of cell division (you don't reference any studies that would suggest there is a global halt to cell division during this stage/condition, and you would clearly have the protocols in place to test for EdU incorporation within the new hindgut in these larval amputations). Without further experimental evidence you cannot conclude that hindgut appearance in larvae at 5 dpa is "morphallaxis" (proliferation-independent).

We absolutely agree that birth of new cells contribute to the appearance of the ciliated hindgut. In fact, the point of Figure 6 is to show birth of new cells by EdU incorporation in the gut epithelium. We think our data strongly supports cell division making an important contribution to tissue remodeling of the digestive system. By removing the term morphallaxis from the text, we hope we have removed any confusion.

Minor points for revision or discussion (future directions, or consider citing prior studies that would add to the current text descriptions):

Lines 156-161: The variability in the number of new segments produced between 3-7 days post-metamorphosis in but cut and uncut animals is interesting. Are these new segments formed sequentially or contemporaneously? Is anything known about what influences this variation? With the data that you have on hand, it would be possible to compare the medians of your cut vs uncut samples to determine whether the medians are significantly different or not (e.g., with a Mann-Whitney U test).

In *Capitella teleta*, new segments are formed from the pgz in a sequential order. This information was added to the introduction to make the paper more accessible to a wider readership. While the exact influences on variation in segment numbers is still an open question, we know from our informal observations that nutrition and overall health can affect segment addition, accounting for some of the variation amongst individuals. We have added a Table containing the segment growth data and performed an analysis of covariance to test for significance.

Is it known whether the vasa and piwi are induced within differentiated cells, that then re-enter the cell cycle to re-form the pgz? The identity of the cells that first express GMP genes, whether this is a homo- or heterogeneous population (e.g., are there ectodermal and mesodermal sources that contribute to the pgz, and are these equivalent in their potential) would be interesting avenues for future study.

We agree with the reviewer that these would be interesting avenues to pursue in the future although they are beyond the scope of the current study. In Boyd and Seaver (2023), we found that amputation induces expression of *vasa* and *piwi* at the wound site. In the same localized area, EdU incorporation indicates cell proliferation, supporting the idea raised by the reviewer that differentiated cells at the wound site are induced to express GMP genes and re-enter the cell cycle. We do not have long term lineage tracers currently available for *Capitella*, which are required to definitively answer this question.

White arrowheads and the amputation plane in the Figure 2 images (nuclei and nervous system panel) are difficult to distinguish from the images and could be bolder and colored so that they stand out.

The arrows in nuclei panels have been changed to the colored pink and made larger. The arrowheads in the nervous system panels were also enlarged.

Line 280: 7 dpm - should this read 7 dpa (days post-amputation)?

This typo has been corrected in the results section of the re-amputation results section.

Your discussion point that onset of regeneration competence may be linked to nutritional sources is interesting, and there is some work in *Xenopus* suggesting that the "refractory" period can be overridden by modulating nutrient stores 10.1016/j.ydbio.2021.01.005.

We appreciate the reviewer bringing this study to our attention. We have incorporated this information by addition of the following statement to the discussion section: "The connection between nutrition and regenerative ability has also been observed in *Xenopus tropicalis* larvae. As tadpoles mature and deplete their yolk stores, their regenerative competence diminishes, yet excess feeding of these tadpoles can prolong regenerative success."

Second decision letter

MS ID#: dev.204995R1

MS TITLE: Regrowing the growth zone: metamorphosis kickstarts regeneration in the annelid *Capitella teleta*

AUTHORS: Alicia A. Boyd and Elaine C. Seaver

Dear Dr Seaver,

I have now received all the referees reports on the above manuscript, and have reached a decision. The referees' comments are appended below.

The overall evaluation is positive and we would like to publish a revised manuscript in Development, provided that the referees' comments can be satisfactorily addressed. Specifically, you should consider whether an ANOVA with post hoc tests would be better for the analyses in Figure 5M, as it alleviates the possibility of false positives caused by multiple t-tests. Alternatively, you might need to consider a multiple testing correction. A clear explanation of which values (error? quartiles?) are indicated by the box and whisker plots is also needed. Further, as pointed out by Reviewer 2, the language (insignificant vs not significant) around statistical results needs editing. Some of the other points raised by Rev 2 could be addressed by showing all samples as dots on the box plot.

Please attend to all of the reviewers' comments in your revised manuscript and detail them in your point-by-point response. If you do not agree with any of their criticisms or suggestions explain clearly why this is so. If it would be helpful, you are welcome to contact us to discuss your revision in greater detail. Please send us a point-by-point response indicating your plans for addressing the referees' comments, and we will look over this and provide further guidance.

Reviewer 1

Advance summary and potential significance to field

I am satisfied with the improvements that the authors made to their already very interesting and well-designed manuscript. There are some issues with the table formatting which I assume only result from pdf conversion, but otherwise I commend the authors for addressing all of the concerns.

Reviewer 2

Advance summary and potential significance to field

As stated in the previous round of review, this work by Boyd and Seaver convincingly shows that posterior regenerative ability, widespread across adult annelids yet shown to be limited/absent during the larval stage in *Capitella teleta*, is restored as individuals of this species undergo metamorphosis into juveniles. This is a clear counter-example to the general perception that regenerative ability decreases as individuals progress through their developmental stages; furthermore, it shows that the regenerative process can be decoupled from the injury event and wound healing process - a prerequisite for its cooption in other developmental strategies, like agametic reproduction or stolonization.

Comments for the author

Having carefully read the authors' responses addressing my previous round of comments and suggestions (and those of the other reviewers), I am mostly satisfied with their changes and replies. My main persisting comment is that I still feel that the handling of statistics throughout the paper is somewhat disappointing. I am fully aware that there is a strong disciplinary bias across developmental biologists, a field where many experiments are quite laborious and often expensive, leading to levels of replication that are below the numbers expected by traditional statistical testing, and where proof-of-concept results are more important than very low p-values. This leads to less intensive training in statistical theory and practice, and to some methodological misconceptions, along with some lag in adopting growing standards in reporting of quantitative analyses and results. However, historical biases should not be a reason to oversee insufficient reporting or erroneous statistical statements in publications. In this manuscript, the authors repeatedly use the term "insignificant" to refer to the results of statistical tests in which the associated p-value was higher than some unspecified alpha-value (I assume the standard 0.05 was used); the correct term is "non-significant" and it means that there was no statistical evidence that the two samples being compared are sampling from two different value distributions. Also, descriptive statistics for such samples should include some measure of variance, as this is even

more important to report than all three estimates of centrality (mean, median and mode). If statistical testing is performed, then the value of the statistic should be stated, along with confidence intervals for the estimates, so a reader can have some idea of variance degree and thus, effect size. A description of the box-and-whisker plot in Fig. 5M is needed in the legend, since not all implementations of the plot use the same descriptors to delimitate the box margins and whisker values. Ideally, the actual data tables would be provided so others can replicate the statistical analyses or perform different ones - and many publications now also include supporting scripts showing exactly how each statistical analysis was performed.

My one other suggestion is that the reader should be reminded here and there that regeneration in this system is posterior regeneration only. With so many annelids also showing strong anterior regeneration abilities, a reader unfamiliar with *Capitella* may not know the fact that this species is unable to regenerate anterior ends, and assume that metamorphosis enables a broader potential than it actually does (and now I am wondering if an anteriorly amputated larva would survive, metamorphose and perhaps make a better attempt to restore a missing anterior end).

All previous comments notwithstanding, I think this manuscript is a great piece of work on its own, an even better follow up to the previous publication showing lack of larval regeneration in the species, and a superb contribution to annelid regeneration and evo-devo. Congratulations to the authors!

Eduardo Zattara

Reviewer 3

Advance summary and potential significance to field

The authors have addressed all major concerns brought up during the first round of revisions. I appreciate that you have undertaken rigorous quantification of the hindgut/total gut length in the revised manuscript. However, I would suggest that someone with expertise in statistics review the comparisons in Figure 5M. These are not strictly pairwise comparisons - some samples are compared to more than one other condition - making t-tests potentially insufficient for the comparisons. Two-way ANOVA and appropriate post-hoc tests may be a better way to compare uncut, JAL, and JAJ samples at one or more time points.

Typos: line 144-145: presence of ?

Line 168: should this time point refer to 14 dpm for the JAL cohort?

Figure 5M: 14 day JAJ (black) - listed as JAL on X axis?

Line 411: "JAL gut regeneration is different from JAL"

Second revision

Author response to reviewers' comments

Note: responses to detailed reviewer comments are in bold.

Reviewer 2: SUGGESTIONS TO AUTHORS

- In this manuscript, the authors repeatedly use the term "insignificant" to refer to the results of statistical tests in which the associated p-value was higher than some unspecified alpha-value (I assume the standard 0.05 was used); the correct term is "non-significant" and it means that there was no statistical evidence that the two samples being compared are sampling from two different value distributions.

All uses of the term "insignificant" have been removed and replaced with the term "non-significant." We also added a phrase stating that the standard of 0.05 was used to Table 2.

- Also, descriptive statistics for such samples should include some measure of variance, as this is even more important to report than all three estimates of centrality (mean, median and mode). If statistical testing is performed, then the value of the statistic should be stated, along with confidence intervals for the estimates, so a reader can have some idea of variance degree and thus, effect size.
We adapted table 1 for new segments to include standard deviation and variance and removed mode and median from the table. We also generated a similar table (table 2) for ciliated gut: whole body length ratios, including average with standard deviation and variance.
- A description of the box-and-whisker plot in Fig. 5M is needed in the legend, since not all implementations of the plot use the same descriptors to delimitate the box margins and whisker values.
The legend for Fig 5M has been moved to Figure 3 and updated to incorporate the requested additional information. The legend now reads:
“Box and whisker plot showing proportion of the body length composed of ciliated hindgut for each experimental group. Y-axis plots the ciliated hindgut: whole body ratio. CG, ciliated gut; WL, whole body length. X-axis includes each experimental group. Uncut and cut animals for each experimental group are represented by the following colors: 3 day uncut juvenile, dark blue; 3-day JAL, red; 3 day JAJ, yellow; uncut stage 9 larvae, green; 3 dpa larvae, purple; 7 day uncut juvenile, light blue; 7-day JAL, pink; 5 day JAJ, light green; 5 dpa larvae, teal; 14 day uncut juvenile, black; and 14-day JAL, grey. JAL, juveniles amputated as larvae; JAJ, juveniles amputated as juveniles. Each value is represented by a dot on the plot, next to the associated box-and-whisker plot in matching color. Within each box, dotted horizontal lines indicate the mean and solid lines indicate the median.”
- Ideally, the actual data tables would be provided so others can replicate the statistical analyses or perform different ones - and many publications now also include supporting scripts showing exactly how each statistical analysis was performed.
The raw length measurements and ratios for each animal’s ciliated gut/whole body length have been compiled into a supplemental table.
- My one other suggestion is that the reader should be reminded here and there that regeneration in this system is posterior regeneration only. With so many annelids also showing strong anterior regeneration abilities, a reader unfamiliar with *Capitella* may not know the fact that this species is unable to regenerate anterior ends, and assume that metamorphosis enables a broader potential than it actually does (and now I am wondering if an anteriorly amputated larva would survive, metamorphose and perhaps make a better attempt to restore a missing anterior end).
We added some varying clarifying phrases, such as “posterior regeneration” or “regeneration of posterior structures” into the text to remind readers that *Capitella teleta* regeneration, and therefore this paper, is restricted to posterior regeneration.

Reviewer 3:

- The authors have addressed all major concerns brought up during the first round of revisions. I appreciate that you have undertaken rigorous quantification of the hindgut/total gut length in the revised manuscript. However, I would suggest that someone with expertise in statistics review the comparisons in Figure 5M. These are not strictly pairwise comparisons - some samples are compared to more than one other condition - making t-tests potentially insufficient for the comparisons. Two-way ANOVA and appropriate post-hoc tests may be a better way to compare uncut, JAL, and JAJ samples at one or more time points.
We consulted a colleague with statistical expertise to discuss the statistics surrounding the hindgut quantitative analyses. We concluded that since we only compared two conditions at a given time, rather than 3 or more (ANOVA), a pairwise comparison is the best fit statistical analysis. We did use control data sets in multiple comparisons, but only with one other condition at a time. We realized that the representation of comparisons on the box-and whisker plot likely contributed to some of this misunderstanding. Therefore, we removed the brackets and asterisks from the plot, reporting the statistical comparisons and results in Table 3.

- Typos: line 144-145: presence of ? **The word 'ganglia' was added to this sentence.**
Line 168: should this time point refer to 14 dpm for the JAL cohort? **Yes, and this has been corrected.**

Figure 5M: 14 day JAJ (black) - listed as JAL on X axis? **A new plot was generated and this has been fixed.**

Line 411: "JAL gut regeneration is different from JAL" **The second JAL has been corrected and replaced with 'JAJ'.**

Third decision letter

MS ID#: dev.204995R2

MS TITLE: Regrowing the growth zone: metamorphosis kickstarts regeneration in the annelid *Capitella teleta*

AUTHORS: Alicia A. Boyd and Elaine C. Seaver

Dear Dr Seaver,

I am happy to tell you that your manuscript has been accepted for publication in Development, pending our standard publication integrity checks.